# Inducing Neural Collapse in Imbalanced Learning: Do We Really Need a Learnable Classifier at the End of Deep Neural Network?

**Yibo Yang**[1], **Shixiang Chen**[1], **Xiangtai Li**[2], **Liang Xie**[3], **Zhouchen Lin**[2,4,5*], **Dacheng Tao**[1,6]

[1]JD Explore Academy, Beijing, China
[2]Key Lab. of Machine Perception (MoE), School of Intelligence Science and Technology, Peking University
[3]State Key Lab of CAD&CG, Zhejiang University
[4]Institute for Artificial Intelligence, Peking University
[5]Pazhou Laboratory, Guangzhou, China
[6]The University of Sydney, Australia

## Abstract

Modern deep neural networks for classification usually jointly learn a backbone for representation and a linear classifier to output the logit of each class. A recent study has shown a phenomenon called *neural collapse* that the within-class means of features and the classifier vectors converge to the vertices of a simplex equiangular tight frame (ETF) at the terminal phase of training on a balanced dataset. Since the ETF geometric structure maximally separates the pair-wise angles of all classes in the classifier, it is natural to raise the question, *why do we spend an effort to learn a classifier when we know its optimal geometric structure?* In this paper, we study the potential of learning a neural network for classification with the classifier randomly initialized as an ETF and fixed during training. Our analytical work based on the layer-peeled model indicates that the feature learning with a fixed ETF classifier naturally leads to the neural collapse state even when the dataset is imbalanced among classes. We further show that in this case the cross entropy (CE) loss is not necessary and can be replaced by a simple squared loss that shares the same global optimality but enjoys a better convergence property. Our experimental results show that our method is able to bring significant improvements with faster convergence on multiple imbalanced datasets.

## 1 Introduction

Modern deep neural networks for classification are composed of a backbone network to extract features, and a linear classifier in the last layer to predict the logit of each class. As widely adopted in various deep learning fields, the linear classifier has been learnable jointly with the backbone network using the cross entropy (CE) loss function for classification problems [19, 10, 13].

A recent study reveals a very symmetric phenomenon named neural collapse, that the last-layer features of the same class will collapse to their within-class mean, and the within-class means of all classes and their corresponding classifier vectors will converge to the vertices of a simplex equiangular tight frame (ETF) at the terminal phase of training on a balanced dataset [28]. A simplex ETF describes a geometric structure that maximally separates the pair-wise angles of $K$ vectors in $\mathbb{R}^d$, $d \geq K - 1$, and all vectors have an equal $\ell_2$ norm. As shown in Figure 1, when $d = K - 1$, the ETF reduces to a regular simplex. Following studies focus on unveiling the physics of such a phenomenon based on the layer-peeled model (LPM) [5] or unconstrained feature model [24]. They peel off the topmost layer, so the features are independent variables to optimize [41]. Although this toy model is impracticable for application, it inherits the nature of feature and classifier learning in real deep networks.

---

*: corresponding author.

It has been shown that the optimality of LPM satisfies neural collapse under the CE loss with constraints [22, 5, 41, 8], regularization [50], or even no explicit constraint [15]. The other studies turn to analyze the mean squared error (MSE) loss and also derive the neural collapse solution as global optimality [24, 9, 31, 38]. However, all these theoretical results are only valid for balanced training.

Albeit neural collapse is an observed empirical result and has not been entirely understood from a theoretical point, it is intuitive and sensible. Features collapsing to their means minimize the within-class variability, while the ETF geometric structure maximizes the between-class variability, so the Fisher discriminant ratio [6, 33] is maximized, which corresponds to an optimal linearly separable state for classification. Several works have shown that neural networks progressively increase the linear separability during training [26, 27, 46]. Therefore, a classifier that satisfies the ETF structure should be a "correct" answer for network training. However, as pointed out by [5], in the training on an imbalanced dataset, the classifier vectors of minor classes will be merged, termed as *minority collapse*, which breaks up the ETF structure and deteriorates the performance on test data. So a learnable classifier does not always lead to neural collapse when training on imbalanced data. With the spirit of *Occam's Razor* [36], we raise and study the question:

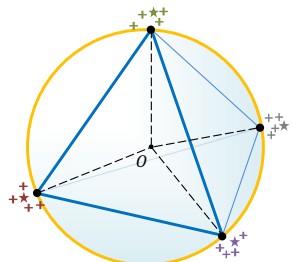

Figure 1: An illustration of a simplex equiangular tight frame when $d = 3$ and $K = 4$. The black spheres are the vertices of the ETF. The "+" and "⋆" signs in different colors refer to features and classifier vectors of different classes, respectively. Neural collapse indicates that the features and classifier vectors are aligned with the same simplex ETF.

*Do we really need to learn a linear classifier at the end of deep neural network?*

Since a backbone network is usually over-parameterized and can align the output feature with any direction, what makes classification effective should be the feature-classifier geometric structure, instead of their specific directions. In this paper, we propose to initialize the classifier using a randomly generated ETF and fix it during training, *i.e.,* only the backbone is learnable. It turns out that this practice makes the LPM more mathematically tractable. Our analytical work indicates that even in the training on an imbalanced dataset, the features will converge to the vertices of the same ETF formed by the fixed classifier, *i.e.,* neural collapse inherently emerges regardless of class balance.

We further analyze the cross entropy loss function. We point out the reason for minority collapse from the perspective of imbalanced gradients with respect to a learnable classifier. As a comparison, our fixed ETF classifier does not suffer from this dilemma. We also show that the gradients with respect to the last-layer feature are composed of a "pull" term that pulls the feature into the direction of its corresponding classifier vector of the same class, and a "push" term that pushes it away from the other classifier vectors. It is the pull-push mechanism in the CE loss that makes the features collapsed and separated. However, in our case, the classifier has been fixed as a "correct" answer. As a result, the "pull" term is always an accurate gradient towards the optimality, and we no longer need the "push" term that may cause inaccurate gradients. Inspired by the analyses, we further propose a simple squared loss, named dot-regression (DR) loss, which shares a similar "pull" gradient with the CE loss, but does not have any "push" term. It has the same neural collapse global optimality, but is proved to enjoy a better convergence property than the CE loss.

The contributions of this study can be listed as follows:

- We theoretically show that the neural collapse optimality can be induced even in imbalanced learning as long as the learnable classifier is fixed as a simplex ETF.

- Our analysis indicates that the broken neural collapse in imbalanced learning is caused by the imbalanced gradients *w.r.t.* a learnable classifier.

- We point out that our fixed ETF classifier no longer relies on the "push" gradient *w.r.t* feature that is crucial for the CE loss but may be inaccurate. We further propose a new loss function with only "pull" gradient. Its better convergence property is theoretically proved.

- Statistical results show that the model with our method converges more closely to neural collapse. In experiments, our method is able to improve the performance of classification on multiple imbalanced datasets by a significant margin. Our method enjoys a faster convergence compared with the traditional learnable classifier with the CE loss. As an extension, we find that our method also works for fine-grained classification.

## 2   Related Work

**Neural Network for Classification.** Despite the success of deep learning for classification [18, 35, 10, 13, 45, 11], a theoretical foundation that can guide the design of neural architecture still remains an open problem and inspires many studies from different perspectives [7, 23, 20, 44, 1, 14, 42]. The last-layer linear classifier has been relatively transparent. In most cases, it is jointly optimized with the backbone network. In long-tailed classification, a two-stage method of training backbone and classifier successively is preferred [2, 16, 48]. Some prior studies have shown that fixing the classifier as regular polytopes [29, 30] and Hadamard matrix [12] does not harm the classification performance. Zhu et al. [50] tried the practice of fixing the classifier as a simplex ETF in experiment, but does not provide any benefit except for the saved computation cost. We also study the potential of using a fixed classifier throughout the training. But different from these studies, our proposed practice is proved to be beneficial to imbalanced learning by both theoretical and experimental results.

**Neural Collapse.** In [28], neural collapse was observed at the terminal phase of training on a balanced dataset. Albeit the phenomenon is intuitive, its reason has not been entirely understood, which inspires several lines of theoretical work on it. Papyan et al. [28] proved that if features satisfy neural collapse, the optimal classifier vectors under the MSE loss will also converge to neural collapse based on [40]. Some studies turn to a simplified model that only considers the last-layer features and classifier as independent variables. They prove that neural collapse emerges under the CE loss with proper constraints or regularizations [41, 5, 22, 50, 15, 8]. Other studies focus on the neural collapse under the MSE loss [24, 9, 31, 38, 49, 32]. In [4], a convex formulation is proposed for a particular network to explain neural collapse. ***However, current results are only valid for balanced training.*** Inspired by neural collapse, the study [43] modifies the CE loss for imbalanced learning, but still cannot rigorously induce neural collapse. Our work differs from these studies in that ***we theoretically show that neural collapse can inherently happen even in imbalanced learning***. We also derive a new loss function that theoretically enjoys a better convergence property than the CE loss.

## 3   Preliminaries

### 3.1   Neural Collapse

Papyan et al. [28] revealed the phenomenon that the last-layer features will converge to their within-class means, and the within-class means together with the classifier vectors will collapse to the vertices of a simplex equiangular tight frame at the terminal phase of training on a balanced dataset.

**Definition 1 (Simplex Equiangular Tight Frame)** *A collection of vectors* $\mathbf{m}_i \in \mathbb{R}^d$, $i = 1, 2, \cdots, K$, $d \geq K - 1$, *is said to be a simplex equiangular tight frame if:*

$$\mathbf{M} = \sqrt{\frac{K}{K-1}} \mathbf{U} \left( \mathbf{I}_K - \frac{1}{K} \mathbf{1}_K \mathbf{1}_K^T \right), \tag{1}$$

*where* $\mathbf{M} = [\mathbf{m}_1, \cdots, \mathbf{m}_K] \in \mathbb{R}^{d \times K}$, $\mathbf{U} \in \mathbb{R}^{d \times K}$ *allows a rotation and satisfies* $\mathbf{U}^T \mathbf{U} = \mathbf{I}_K$, $\mathbf{I}_K$ *is the identity matrix, and* $\mathbf{1}_K$ *is an all-ones vector.*

All vectors in a simplex ETF have an equal $\ell_2$ norm and the same pair-wise angle, *i.e.*,

$$\mathbf{m}_i^T \mathbf{m}_j = \frac{K}{K-1} \delta_{i,j} - \frac{1}{K-1}, \forall i, j \in [1, K], \tag{2}$$

where $\delta_{i,j}$ equals to 1 when $i = j$ and 0 otherwise. The pair-wise angle $-\frac{1}{K-1}$ is the maximal equiangular separation of $K$ vectors in $\mathbb{R}^d$ [28].

Then the neural collapse (NC) phenomenon can be formally described as:

**(NC1)** Within-class variability of the last-layer features collapse: $\Sigma_W \rightarrow \mathbf{0}$, and $\Sigma_W := \mathrm{Avg}_{i,k}\{(\mathbf{h}_{k,i} - \mathbf{h}_k)(\mathbf{h}_{k,i} - \mathbf{h}_k)^T\}$, where $\mathbf{h}_{k,i}$ is the last-layer feature of the $i$-th sample in the $k$-th class, and $\mathbf{h}_k = \mathrm{Avg}_i\{\mathbf{h}_{k,i}\}$ is the within-class mean of the last-layer features in the $k$-th class;

**(NC2)** Convergence to a simplex ETF: $\tilde{\mathbf{h}}_k = (\mathbf{h}_k - \mathbf{h}_G)/||\mathbf{h}_k - \mathbf{h}_G||, k \in [1, K]$, satisfies Eq. (2), where $\mathbf{h}_G$ is the global mean of the last-layer features, *i.e.*, $\mathbf{h}_G = \mathrm{Avg}_{i,k}\{\mathbf{h}_{k,i}\}$;

**(NC3)** Self duality: $\tilde{\mathbf{h}}_k = \mathbf{w}_k/||\mathbf{w}_k||$, where $\mathbf{w}_k$ is the classifier vector of the $k$-th class;

**(NC4)** Simplification to the nearest class center prediction: $\mathrm{argmax}_k\langle\mathbf{h}, \mathbf{w}_k\rangle = \mathrm{argmin}_k||\mathbf{h} - \mathbf{h}_k||$, where $\mathbf{h}$ is the last-layer feature of a sample to predict for classification.

## 3.2 Layer-peeled Model

Neural collapse has attracted a lot of researchers to unveil the physics of such an elegant phenomenon. Currently most studies target the cross entropy loss function that is widely used in deep learning for classification. It is defined as:

$$\mathcal{L}_{CE}(\mathbf{h}, \mathbf{W}) = -\log\left(\frac{\exp(\mathbf{h}^T\mathbf{w}_c)}{\sum_{k=1}^{K}\exp(\mathbf{h}^T\mathbf{w}_k)}\right),$$ (3)

where $\mathbf{h} \in \mathbb{R}^d$ is the feature output by a backbone network with input $\mathbf{x}$, $\mathbf{W} = [\mathbf{w}_1, \cdots, \mathbf{w}_K] \in \mathbb{R}^{d \times K}$ is a learnable classifier, and $c$ is the class label of $\mathbf{x}$.

However, deep neural network as a highly interacted function is difficult to analyze due to its non-convexity. A simplification is always necessary to make tractable analysis. For neural collapse, current studies often consider the case where only the last-layer features and classifier are learnable without considering the layers in the backbone network. It is termed as layer-peeled model (LPM) [5], and can be formulated as[1]:

$$\min_{\mathbf{W},\mathbf{H}} \quad \frac{1}{N}\sum_{k=1}^{K}\sum_{i=1}^{n_k}\mathcal{L}_{CE}(\mathbf{h}_{k,i}, \mathbf{W}),$$
$$s.t. \quad ||\mathbf{w}_k||^2 \le E_W, \ \forall 1 \le k \le K,$$
$$||\mathbf{h}_{k,i}||^2 \le E_H, \ \forall 1 \le k \le K, 1 \le i \le n_k,$$ (4)

where $\mathbf{h}_{k,i}$ is the feature of the $i$-th sample in the $k$-th class, $\mathbf{W}$ is a learnable classifier, $\mathcal{L}_{CE}$ is the cross entropy loss function defined in Eq. (3), $N$ is the number of samples, and $E_H$ and $E_W$ are the $\ell_2$ norm constraints for feature $\mathbf{h}$ and classifier vector $\mathbf{w}$, respectively.

Albeit the LPM cannot be applied to real application problems, it serves as an analytical tool and inherits the learning behaviors of the last-layer features and classifier in deep neural network. Actually, the learning of a backbone network is through the multiplication between the Jacobian and the gradient with respect to the last-layer features, *i.e.*, $\frac{\partial \mathbf{H}}{\partial \mathbf{W}_{1:L-1}}\frac{\partial \mathcal{L}}{\partial \mathbf{H}}$, where $\mathbf{W}_{1:L-1}$ denotes the parameters in the backbone network, and $\mathbf{H}$ is the collection of the last-layer features.

It has been shown that the global optimality for the LPM in Eq. (4) satisfies neural collapse in the balanced case [5, 8]. The CE loss with regularization also has a similar conclusion [50]. *However, the results in current studies are only valid for training on a balanced dataset, i.e.,* $n_k = n, \forall 1 \le k \le K$.

# 4 Main Results

## 4.1 ETF Classifier

From the neural collapse solution, NC1 minimizes the within-class covariance $\Sigma_W$, and NC2 maximizes the between-class covariance $\Sigma_B$ by the ETF structure. So the Fisher discriminant ratio, defined as $\Sigma_W^{-1}\Sigma_B$, is maximized, which can measure the linear separability and has been used to extract features to replace the CE loss [37]. So we deem an ETF as the optimal geometric structure for the linear classifier. Considering that a backbone network is usually over-parameterized and can produce features aligned with any direction, in this paper, we study the potential of learning a network with the linear classifier fixed as an ETF, named ETF classifier.

Concretely, we initialize the linear classifier $\mathbf{W}$ as a random simplex ETF by Eq. (1) with a scaling of $\sqrt{E_W}$ as the fixed length for each classifier vector, and only optimize the features $\mathbf{H}$. In this case, the layer-peeled model (LPM) in Eq. (4) reduces to the following problem:

$$\min_{\mathbf{H}} \quad \frac{1}{N}\sum_{k=1}^{K}\sum_{i=1}^{n_k}\mathcal{L}_{CE}(\mathbf{h}_{k,i}, \mathbf{W}^*),$$
$$s.t. \quad ||\mathbf{h}_{k,i}||^2 \le E_H, \ \forall 1 \le k \le K, 1 \le i \le n_k,$$ (5)

---

[1]Note that the sample-wise constraint of $\mathbf{H}$ and the class-wise constraint of $\mathbf{W}$ in Eq. (4) are more strict than the overall constraints in [5], but are still active with the same global optimality. The model is also known as unconstrained feature model [24, 50] when the norm constraints are omitted or replaced by regularizations.

where $\mathbf{W}^*$ is the fixed classifier as a simplex ETF and satisfies:

$$\mathbf{w}_k^{*T}\mathbf{w}_{k'}^* = E_W\left(\frac{K}{K-1}\delta_{k,k'} - \frac{1}{K-1}\right), \forall k, k' \in [1, K], \tag{6}$$

where $\delta_{k,k'}$ equals to 1 when $k = k'$ and 0 otherwise.

We observe that this practice decouples the multiplied learnable variables $\mathbf{h}_{k,i}$ and $\mathbf{w}_k$ of LPM in Eq. (4), and makes the model in Eq. (5) a convex problem that is more mathematically tractable. We term the decoupled LPM in Eq. (5) as **DLPM** for short. We have the global optimality for DLPM in the imbalanced case with the ETF classifier in the following theorem.

**Theorem 1** *No matter the data distribution is balanced or not among classes (it is allowed that $\exists k, k' \in [1, K]$, $k \neq k'$, such that $n_k \gg n_{k'}$), any global minimizer $\mathbf{H}^* = [\mathbf{h}_{k,i}^* : 1 \leq k \leq K, 1 \leq i \leq n_k]$ of Eq. (5) converges to a simplex ETF with the same direction as $\mathbf{W}^*$ and a length of $\sqrt{E_H}$, i.e.,*

$$\mathbf{h}_{k,i}^{*T}\mathbf{w}_{k'}^* = \sqrt{E_H E_W}\left(\frac{K}{K-1}\delta_{k,k'} - \frac{1}{K-1}\right), \forall\, 1 \leq k, k' \leq K, 1 \leq i \leq n_k, \tag{7}$$

*which means that the neural collapse phenomenon emerges regardless of class balance.*

**Proof 1** *Please refer to Appendix A for our proof.* □

**Remark 1** *As observed in [5], LPM in the extreme imbalance case would suffer from "minority collapse", where the classifier vectors of minor classes are close or even merged into the same vector, which explains the deteriorated classification performance of imbalanced training. As a comparison, Theorem 1 shows that DLPM with our ETF classifier can inherently produce the neural collapse solution even in the training on imbalanced data.*

Although our practice of using a fixed ETF classifier simplifies the problem, it actually brings theoretical merits that also get validated in our long-tailed classification experiments.

## 4.2 Rethinking the Cross Entropy Loss

In this subsection, we rethink the CE loss $\mathcal{L}_{CE}$ from the perspective of gradients with respect to both feature and classifier to analyze its learning behaviors.

### 4.2.1 Gradient *w.r.t* Classifier

We first analyze the gradient of $\mathcal{L}_{CE}$ *w.r.t* a learnable classifier $\mathbf{W} = [\mathbf{w}_1, \cdots, \mathbf{w}_K] \in \mathbb{R}^{d \times K}$:

$$\frac{\partial \mathcal{L}_{CE}}{\partial \mathbf{w}_k} = \sum_{i=1}^{n_k} -\left(1 - p_k\left(\mathbf{h}_{k,i}\right)\right)\mathbf{h}_{k,i} + \sum_{k' \neq k}^{K} \sum_{j=1}^{n_{k'}} p_k\left(\mathbf{h}_{k',j}\right)\mathbf{h}_{k',j}, \tag{8}$$

where $p_k(\mathbf{h})$ is the predicted probability that $\mathbf{h}$ belongs to the $k$-th class. It is calculated by the softmax function and takes the following form in the CE loss:

$$p_k(\mathbf{h}) = \frac{\exp(\mathbf{h}^T\mathbf{w}_k)}{\sum_{k'=1}^{K}\exp(\mathbf{h}^T\mathbf{w}_{k'})}, \ 1 \leq k \leq K. \tag{9}$$

We make the following definitions:

$$-\frac{\partial \mathcal{L}_{CE}}{\partial \mathbf{w}_k} = \mathbf{F}_{\text{pull}}^{(\mathbf{w})} + \mathbf{F}_{\text{push}}^{(\mathbf{w})}, \tag{10}$$

where

$$\mathbf{F}_{\text{pull}}^{(\mathbf{w})} = \sum_{i=1}^{n_k}\left(1 - p_k\left(\mathbf{h}_{k,i}\right)\right)\mathbf{h}_{k,i}, \quad \mathbf{F}_{\text{push}}^{(\mathbf{w})} = -\sum_{k' \neq k}^{K}\sum_{j=1}^{n_{k'}} p_k\left(\mathbf{h}_{k',j}\right)\mathbf{h}_{k',j}. \tag{11}$$

It reveals that the negative gradient *w.r.t* $\mathbf{w}_k$ is decomposed into two terms. The "pull" term $\mathbf{F}_{\text{pull}}^{(\mathbf{w})}$ pulls $\mathbf{w}_k$ towards the directions of the features of the same class, *i.e.*, $\mathbf{h}_{k,i}$, while the "push" term $\mathbf{F}_{\text{push}}^{(\mathbf{w})}$ pushes $\mathbf{w}_k$ away from the directions of the features of the other classes, *i.e.*, $\mathbf{h}_{k',i}, \forall k' \neq k$.

**Remark 2** *Note that* $\mathbf{F}^{(\mathbf{w})}_{\text{pull}}$ *and* $\mathbf{F}^{(\mathbf{w})}_{\text{push}}$ *have different magnitudes. When the dataset is in extreme imbalance, the direction of* $-\frac{\partial \mathcal{L}_{CE}}{\partial \mathbf{w}_k}$ *for a minor class is dominated by the push term* $\mathbf{F}^{(\mathbf{w})}_{\text{push}}$ *because* $n_k$ *is small, while* $N - n_k$ *is large. In this case, the classifier vectors of two minor classes are optimized towards nearly the same direction and would be close or merged after training. So the deteriorated performance of classification with imbalanced training data results from the imbalanced gradient w.r.t a learnable classifier in the CE loss. As a comparison, our proposed practice of fixing the classifier as an ETF does not suffer from this dilemma.*

#### 4.2.2 Gradient *w.r.t* Feature

The gradient of CE loss in Eq. (3) with respect to $\mathbf{h}$ is:

$$\frac{\partial \mathcal{L}_{CE}}{\partial \mathbf{h}} = - \left( 1 - p_c(\mathbf{h}) \right) \mathbf{w}_c + \sum_{k \neq c}^{K} p_k(\mathbf{h}) \mathbf{w}_k, \tag{12}$$

where $c$ is the class label of $\mathbf{h}$, and $p_k(\mathbf{h})$ is the probability that $\mathbf{h}$ belongs to the $k$-th class as defined in Eq. (9).

It is shown that Eq. (12) has a similar form to that of Eq. (8). The negative gradient $-\frac{\partial \mathcal{L}_{CE}}{\partial \mathbf{h}}$ can also be decomposed as the addition of "pull" and "push" terms defined as:

$$\mathbf{F}^{(\mathbf{h})}_{\text{pull}} = \left( 1 - p_c(\mathbf{h}) \right) \mathbf{w}_c, \quad \mathbf{F}^{(\mathbf{h})}_{\text{push}} = - \sum_{k \neq c}^{K} p_k(\mathbf{h}) \mathbf{w}_k. \tag{13}$$

The "pull" term $\mathbf{F}^{(\mathbf{h})}_{\text{pull}}$ pulls $\mathbf{h}$ towards the classifier vector of the same class, *i.e.*, $\mathbf{w}_c$, while the "push" term $\mathbf{F}^{(\mathbf{h})}_{\text{push}}$ pushes $\mathbf{h}$ away from the other classifier vectors, *i.e.*, $\mathbf{w}_k, \forall k \neq c$.

**Remark 3** *We note that Eq. (11) and Eq. (13) have a similar form and are very symmetric. The "pull" terms* $\mathbf{F}^{(\mathbf{w})}_{\text{pull}}$ *and* $\mathbf{F}^{(\mathbf{h})}_{\text{pull}}$ *force* $\mathbf{w}$ *and* $\mathbf{h}$ *of the same class to converge to the same direction, which is in line with (NC1) and (NC3). The "push" terms* $\mathbf{F}^{(\mathbf{w})}_{\text{push}}$ *and* $\mathbf{F}^{(\mathbf{h})}_{\text{push}}$ *make them of different classes separated, which is in line with (NC2). Generally, (NC1)-(NC3) can lead to (NC4). We remark that it is the "pull-push" mechanism in the CE loss that leads to the neural collapse solution in the case of training on balanced data.*

The "push" gradient $\mathbf{F}^{(\mathbf{h})}_{\text{push}}$ in Eq. (13) is crucial for a learnable classifier. However, in our case where the classifier has been fixed as an ETF, the "push" gradient $\mathbf{F}^{(\mathbf{h})}_{\text{push}}$ is unnecessary. As shown in Figure 2, the "pull" gradient $\mathbf{F}^{(\mathbf{h})}_{\text{pull}}$ in our case is always accurate towards the optimality $\mathbf{w}^*_c$, which corresponds to the neural collapse solution. But the "push" gradient does not necessarily direct to the optimality. So, we no longer rely on the "push" term $\mathbf{F}^{(\mathbf{h})}_{\text{push}}$ that may cause deviation. It inspires us to develop a new loss function specified for our ETF classifier.

### 4.3 Dot-Regression Loss

We consider the following squared loss function:

$$\mathcal{L}_{DR}(\mathbf{h}, \mathbf{W}^*) = \frac{1}{2\sqrt{E_W E_H}} \left( \mathbf{w}^{*T}_c \mathbf{h} - \sqrt{E_W E_H} \right)^2, \tag{14}$$

where $c$ is the class label of $\mathbf{h}$, $\mathbf{W}^*$ is a fixed ETF classifier, and $E_W$ and $E_H$ are the $\ell_2$-norm constraints (predefined and *not learnable*) given in Eq. (5). It performs regression between the dot product of $\mathbf{h}$ and $\mathbf{w}^*_c$ and the multiplication of their lengths. We term this simple loss function as dot-regression (DR) loss. Its gradient with respect to $\mathbf{h}$ takes the form:

$$\frac{\partial \mathcal{L}_{DR}}{\partial \mathbf{h}} = - \left( 1 - \cos \angle(\mathbf{h}, \mathbf{w}^*_c) \right) \mathbf{w}^*_c, \tag{15}$$

where $\cos \angle(\mathbf{h}, \mathbf{w}^*_c)$ denotes the cosine similarity between $\mathbf{h}$ and $\mathbf{w}^*_c$.

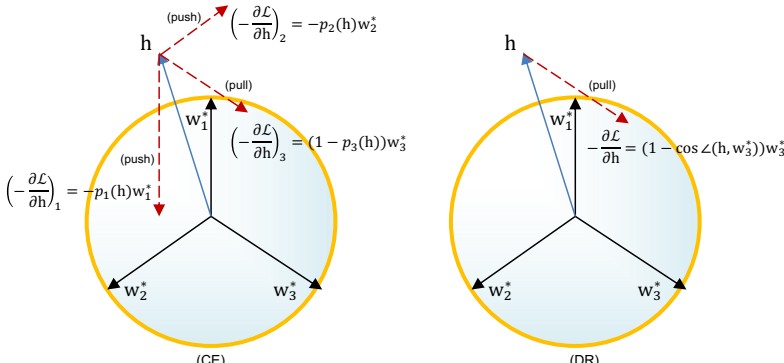

Figure 2: An empirical comparison of gradient directions *w.r.t* an $\mathbf{h}$ (belongs to the 3-rd class) of the CE loss (left) and our proposed DR loss (right). Because $\mathbf{h}$ is close to $\mathbf{w}_1^*$, the gradient of the CE loss is dominated by $-\left(\frac{\partial \mathcal{L}_{CE}}{\partial \mathbf{h}}\right)_1$ whose direction deviates from the optimality $\mathbf{w}_3^*$, while the gradient of DR only has a "pull" term and always directs to $\mathbf{w}_3^*$.

We see that the gradient has a similar form to the first term in Eq. (12), which plays the role of "pull", but has no "push" term. It is easy to identify that if we replace the CE loss in the decoupled layer-peeled model (DLPM) defined in Eq. (5) with the DR loss defined in Eq. (14), the same global minimizer Eq. (7) still holds. The global optimality happens when $\mathcal{L}_{DR}(\mathbf{h}^*, \mathbf{W}^*) = 0$ and $\cos \angle(\mathbf{h}^*, \mathbf{w}_c^*) = 1$ accordingly. Then we give a formal analysis of the convergence properties of both CE and DR loss functions in the DLPM.

**Definition 2** *Given $\delta > 0$, for any $\mathbf{h}$ satisfying $||\mathbf{h} - \mathbf{h}^*|| \leq \delta$ and $||\mathbf{h}||^2 = E_H$, the $\eta-$regularity number of function $\mathcal{L}(\mathbf{h})$ is defined by the convergence rate of the projected gradient method. That is, there exists $\eta_{\mathbf{h}} \in [0, 1]$ such that:*

$$\| \operatorname{Proj}_{E_H}(\mathbf{h} - \gamma \frac{\partial \mathcal{L}}{\partial \mathbf{h}}) - \mathbf{h}^*\|^2 \leq \eta_{\mathbf{h}}\|\mathbf{h} - \mathbf{h}^*\|^2,$$

*where $\gamma$ is the learning rate such that $\eta_{\mathbf{h}}$ is as small as possible.*

Note that $\eta_{\mathbf{h}}$ is decided by $\mathbf{h}$. For many problems, we cannot find its uniform upper bound $\eta_{\mathbf{h}} \leq \bar{\eta} < 1$. The smaller $\eta_{\mathbf{h}}$ is, the better property the loss function has.

**Theorem 2** *Assume that given a small $\delta > 0$, when $||\mathbf{h} - \mathbf{h}^*|| \leq \delta$, $p(\mathbf{h})$ defined in Eq. (9) satisfies that[2] $p_k(\mathbf{h}) = (1 - p_c(\mathbf{h}))/(K - 1)$, $\forall k \neq c$, where $c$ is the label of $\mathbf{h}$. When optimizing the DLPM defined in Eq. (5) with the CE and DR loss funcitons, for any fixed learning rate $\gamma$, we have:*

$$\eta_{\mathbf{h}}^{(CE)} \geq \frac{1 + \cos \angle(\mathbf{h}, \mathbf{w}_c^*)}{2} = \eta_{\mathbf{h}}^{(DR)}, \tag{16}$$

*where $\eta_{\mathbf{h}}^{(CE)}$ and $\eta_{\mathbf{h}}^{(DR)}$ are the $\eta$-regularity numbers of the CE and DR loss functions, respectively.*

**Proof 2** *Please refer to Appendix B for our proof.* ☐

Eq. (16) indicates that the DR loss has a better convergence property when $\mathbf{h}$ is close to $\mathbf{h}^*$. In implementations, we train a backbone network with our ETF classifier and DR loss. To induce balanced gradients *w.r.t* the backbone network parameters, we define the length of each classifier vector according to class distribution. Please refer to Appendix C for the complete implementation details. In experiments, we also compare our method with the CE loss weighted by class distribution.

## 5 Experiments

In experiments, we first make empirical observations of neural collapse convergence in the imbalanced training with and without our method, and then compare the performances on long-tailed classification. As an extension, we surprisingly find that our method is also able to improve the performance of fine-grained classification, which can be deemed as another imbalanced problem where a majority of features are close to each other. The datasets and training details are described in Appendix D.

---

[2]Its rational lies in that $\mathbf{h}^*$ aligned with $\mathbf{w}_c^*$ has equal dot products with $\mathbf{w}_k^*$, $\forall k \neq c$, and $\mathbf{h}$ is close to $\mathbf{h}^*$.

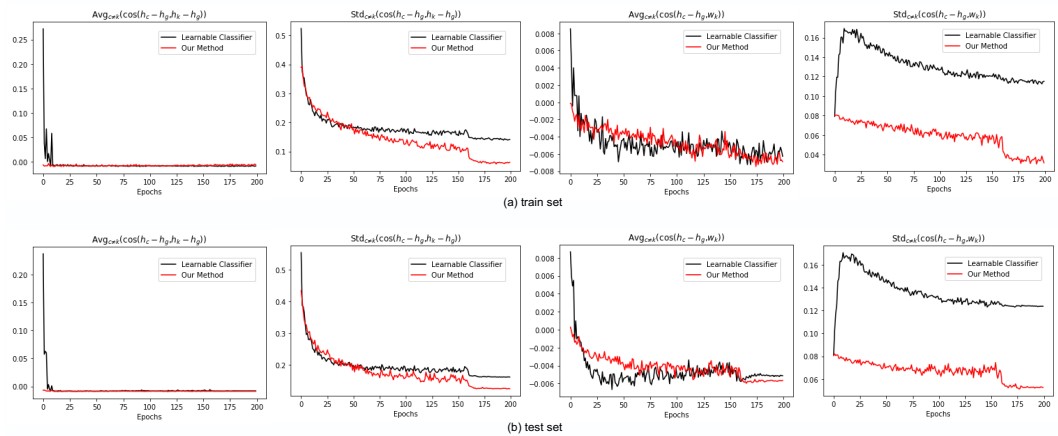

Figure 3: Averages and standard deviations of $\cos\angle(\mathbf{h}_c - \mathbf{h}_g, \mathbf{h}_k - \mathbf{h}_g)$ (two columns on the left), and $\cos\angle(\mathbf{h}_c - \mathbf{h}_g, \mathbf{w}_k)$ (two columns on the right) with (red) and without (black) our method on train set (a) and test set (b). $\mathbf{h}_g$ is the global mean. $\mathbf{h}_c$, $\mathbf{h}_k$ are within-class means, where $c$ and $k$ are all pairs of **different** classes. The models are trained on CIFAR-100 with the imbalance ratio $\frac{n_{\min}}{n_{\max}}$ of 0.02, where $n_{\min}$ and $n_{\max}$ are the minimal and maximal numbers of training samples in all classes.

Table 1: An ablation study with ResNet on CIFAR-10 [17] using different classifiers and loss functions. The numbers in the second row denote the imbalance ratio $\tau = \frac{n_{\min}}{n_{\max}}$, where $n_{\min}$ and $n_{\max}$ are the minimal and maximal numbers of training samples in all classes. Results are the mean of three repeated experiments with different seeds. * indicates that the CE loss is weighted by $\frac{1}{Kn_k}$ to induce class balance, where $K$ is the number of classes and $n_k$ is the number of samples in class $k$.

| Methods | without Mixup | | | | | with Mixup | | | | |
|---|---|---|---|---|---|---|---|---|---|---|
| | 0.005 | 0.01 | 0.02 | 0.1 | balanced | 0.005 | 0.01 | 0.02 | 0.1 | balanced |
| **Learnable Classifier + CE** | $66.1_{\pm 0.3}$ | $71.0_{\pm 0.2}$ | $77.1_{\pm 0.2}$ | $87.4_{\pm 0.2}$ | $93.4_{\pm 0.1}$ | $67.3_{\pm 0.4}$ | $72.8_{\pm 0.3}$ | $78.6_{\pm 0.2}$ | $87.7_{\pm 0.1}$ | $93.6_{\pm 0.2}$ |
| **Learnable Classifier + CE\*** | $66.8_{\pm 0.4}$ | $72.1_{\pm 0.3}$ | $77.6_{\pm 0.3}$ | $87.4_{\pm 0.3}$ | $93.1_{\pm 0.2}$ | $68.5_{\pm 0.3}$ | $73.9_{\pm 0.3}$ | $79.3_{\pm 0.2}$ | $87.8_{\pm 0.2}$ | $93.2_{\pm 0.3}$ |
| **ETF Classifier + CE** | $60.4_{\pm 0.3}$ | $72.9_{\pm 0.3}$ | $79.5_{\pm 0.2}$ | $87.2_{\pm 0.1}$ | $92.6_{\pm 0.2}$ | $60.6_{\pm 0.5}$ | $67.0_{\pm 0.4}$ | $77.2_{\pm 0.3}$ | $87.0_{\pm 0.2}$ | $93.3_{\pm 0.2}$ |
| **ETF Classifier + DR** | $68.4_{\pm 0.2}$ | $73.0_{\pm 0.2}$ | $78.4_{\pm 0.3}$ | $86.9_{\pm 0.2}$ | $92.9_{\pm 0.1}$ | $71.9_{\pm 0.3}$ | $76.5_{\pm 0.3}$ | $81.0_{\pm 0.2}$ | $87.7_{\pm 0.2}$ | $92.0_{\pm 0.2}$ |

## 5.1 Empirical Results

Following [28], we calculate statistics during training to show the neural collapse convergence. We first compare the averages and standard deviations of two cosine similarities, $\cos\angle(\mathbf{h}_c - \mathbf{h}_g, \mathbf{h}_k - \mathbf{h}_g)$ and $\cos\angle(\mathbf{h}_c - \mathbf{h}_g, \mathbf{w}_k)$, where $\mathbf{h}_g$ is the global mean, for all pairs of different classes $(c, k)$, $c \neq k$, with and without our method. As shown in Figure 3, the averages of both $\cos\angle(\mathbf{h}_c - \mathbf{h}_g, \mathbf{h}_k - \mathbf{h}_g)$ and $\cos\angle(\mathbf{h}_c - \mathbf{h}_g, \mathbf{w}_k)$ converge to a negative value near zero. It is consistent with neural collapse that the feature means or classifier vectors of different classes should have a cosine similarity of $-\frac{1}{K-1}$. However, their standard deviations are much smaller when our method is used. It indicates that the models with our method converge to neural collapse more closely.

We further calculate the averages of $\cos\angle(\mathbf{h}_c - \mathbf{h}_g, \mathbf{w}_c)$, $\forall 1 \leq c \leq K$, and $||\tilde{\mathbf{W}} - \tilde{\mathbf{H}}||_F^2$ in Figure 4 and 5 in Appendix E. It reveals that the model using our method generally has a higher $\cos\angle(\mathbf{h}_c - \mathbf{h}_g, \mathbf{w}_c)$ and a lower $||\tilde{\mathbf{W}} - \tilde{\mathbf{H}}||_F^2$, which indicates that the feature means and classifier vectors of the same class are better aligned. We observe no advantage of ResNet on STL-10 and DenseNet on CIFAR-100 in Figure 4 and 5. In Table 2, we see that the two cases are right the failure cases, which shows consistency between neural collapse convergence and classification performance.

## 5.2 Performances on Long-tailed Classification

We conduct an ablation study with ResNet on CIFAR-10. As shown in Table 1, when we replace the learnable classifier with our fixed ETF classifier, the performances get improved for the imbalance ratio $\tau$ of 0.01 and 0.02 without Mixup [47]. They also achieve comparable results for $\tau = 0.1$ and the balanced setting ($\tau = 1$). However, only using the ETF classifier does not work for the extreme imbalance case where $\tau = 0.005$. Besides, it is not compatible with Mixup, which is a

Table 2: Long-tailed classification accuracy (%) with ResNet and DenseNet on four datasets. Results are the mean of three repeated experiments with different seeds.

| Methods | CIFAR-10 [17] | | | CIFAR-100 [17] | | | SVHN [25] | | | STL-10 [3] | | |
|---|---|---|---|---|---|---|---|---|---|---|---|---|
| | 0.005 | 0.01 | 0.02 | 0.005 | 0.01 | 0.02 | 0.005 | 0.01 | 0.02 | 0.005 | 0.01 | 0.02 |
| *ResNet* | | | | | | | | | | | | |
| Learnable Classifier + CE | 67.3 | 72.8 | 78.6 | 38.7 | 43.0 | 48.1 | 40.5 | 40.9 | 49.3 | 33.1 | 37.9 | 38.8 |
| ETF Classifier + DR | 71.9 | 76.5 | 81.0 | 40.9 | 45.3 | 50.4 | 42.8 | 45.7 | 49.8 | 33.5 | 37.2 | 37.9 |
| Improvements | **+4.6** | **+3.7** | **+2.4** | **+2.2** | **+2.3** | **+2.3** | **+2.3** | **+4.8** | **+0.5** | **+0.4** | -0.7 | -0.9 |
| *DenseNet* | | | | | | | | | | | | |
| Learnable Classifier + CE | 71.1 | 77.7 | 84.1 | 40.3 | 43.8 | 49.8 | 39.7 | 40.5 | 46.4 | 38.5 | 41.2 | 44.9 |
| ETF Classifier + DR | 72.9 | 78.5 | 83.4 | 40.1 | 44.0 | 49.7 | 40.5 | 44.8 | 48.4 | 39.5 | 42.9 | 46.3 |
| Improvements | **+1.8** | **+0.8** | -0.7 | -0.2 | **+0.2** | -0.1 | **+0.8** | **+4.3** | **+2.0** | **+1.0** | **+1.7** | **+1.4** |

Table 3: Long-tailed classification accuracy (%) on ImageNet-LT [21] with ResNet-50 backbone and different training epochs.

| Epoch | Methods | Acc. (%) |
|---|---|---|
| 90 | Learnable Classifier + CE | 34.6 |
| | ETF Classifier + DR | 41.8 |
| 120 | Learnable Classifier + CE | 41.9 |
| | ETF Classifier + DR | 43.2 |
| 150 | Learnable Classifier + CE | 42.5 |
| | ETF Classifier + DR | 43.8 |
| 180 | Learnable Classifier + CE | 44.3 |
| | ETF Classifier + DR | 44.7 |

Table 4: Fine-grained classification accuracy (%) on CUB-200-2011 [39] with different ResNet backbones pre-trained on ImageNet. Training details are described in Appendix D.

| Backbone | Methods | Acc. (%) |
|---|---|---|
| ResNet-34 | Learnable Classifier + CE | 82.2 |
| | ETF Classifier + DR | 83.0 |
| ResNet-50 | Learnable Classifier + CE | 85.5 |
| | ETF Classifier + DR | 86.1 |
| ResNet-101 | Learnable Classifier + CE | 86.2 |
| | ETF Classifier + DR | 87.0 |

strong augmentation tool to alleviate the bias brought by adversarial training samples. When the DR loss that is specifically designed for the ETF classifier is used, we achieve significant performance improvements for $\tau$ of 0.005, 0.01, and 0.02 both with and without Mixup. Generally our proposed ETF classifier with the DR loss has better performances on multiple long-tailed cases than the learnable classifier with the original or weighted CE loss.

In Table 2, the advantage of our method is further verified on more datasets with ResNet and DenseNet. Concretely, we achieve an average improvement of 2.0% for ResNet and 1.1% for DenseNet. We also test our method on the large-scale dataset, ImageNet-LT [21], which is an imbalanced version of the ImageNet dataset [34]. As shown in Table 3, we compare our method with baseline by training the models for different epochs. We observe that the superiority of our method is more remarkable when training for less epochs. It can be explained by the fact that our method directly has the classifier in its optimality and optimizes the features towards the neural collapse solution, while the learnable classifier with the CE loss needs a sufficient training process to separate classifier vectors of different classes. So our method can be preferred when fast convergence or limited training time is required. The accuracy curves in training for the results in Table 3 are shown in Figure 6 in Appendix E.

### 5.3 Performances on Fine-grained Classification

Fine-grained classification can also be deemed as an imbalanced problem as the features of multiple classes are close to each other. We surprisingly find that our method is also helpful for fine-grained classification even though most of the analytical work is conducted under the case of class imbalance. As shown in Table 4, our method achieves 0.7%-0.8% accuracy improvements on CUB-200-2011.

## 6 Conclusion

In this paper, we study the potential of training a network with the last-layer linear classifier randomly initialized as a simplex ETF and fixed during training. This practice enjoys theoretical merits under the layer-peeled analytical framework. We further develop a simple loss function specifically for the ETF classifier. Its advantage gets verified by both theoretical and experimental results. We conclude that it is not necessary to learn a linear classifier for classification networks, and our simplified practice even helps to improve the long-tailed and fine-grained classification performances with no cost. Our work may help to further understand neural collapse and the neural network architecture for classification. Code address: https://github.com/NeuralCollapseApplications/ImbalancedLearning.

Limitations and societal impacts are discussed in Appendix F. As suggested by a reviewer of this paper, we compare with [5, 30, 50] in more details in Appendix G.

## Acknowledgments and Disclosure of Funding

Z. Lin was supported by the NSF China (No.s 62276004 and 61731018), NSFC Tianyuan Fund for Mathematics (No. 12026606), and Project 2020BD006 supported by PKU-Baidu Fund.

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
