# OpenReview forum: "Inducing Neural Collapse in Imbalanced Learning: Do We Really Need a Learnable Classifier at the End of Deep Neural Network?"
_NeurIPS.cc/2022/Conference — NeurIPS 2022 Accept_

### Official Review · Reviewer_MaCE · 2022-07-10

**Rating:** 3
**Confidence:** 5
**Soundness:** 2 fair
**Presentation:** 3 good
**Contribution:** 1 poor

**Summary:**

The paper proposes a loss function which projects the features to a pre-defined simplex and computes the gradients w.r.t it. It claims that neural networks use linear classifiers at the end and this is not needed if features are computed for each class and projected to a simplex. A theoretical explanation for this approach is provided and experimental results are presented.

**Questions:**

Line 23:
The paper starts by citing a reference [28] where a phenomenon called neural collapse is mentioned. For example, it is not clear why would features collapse to within-class mean? What is the loss function being used to do so? Are class labels even used during training which leads to this equiangular tight frame? The reader needs to be given a high level/intuitive introduction to ETF at this stage of the paper even though these details may be described in [28]. It could be like, once a neural net is trained, [28] observes 4 properties NC 1,2,3,4, which are the properties for an ETF and we can describe what is NC 1/2/3/4.

Please clarify how the work is different from the papers which I mentioned or if it would be better compared to these in a given scenario.

**Ethics Review Area:**

["I don’t know"]

**Limitations:**

yes, it has been addressed.

**Strengths And Weaknesses:**

The idea to try to solve classification using a simplex is interesting but I believe there are several papers in the deep learning community for large-scale classification problems (like face/few-shot recognition) which have evaluated similar loss functions which have not been cited or compared with in this paper. I am mentioning a few of these papers and they use similar kinds of projection functions but also present more variants of the version presented in this paper. For example, Wen et. al. (ECCV 2016) and Snell et. al. Neurips 2017, use class means/prototypes as a loss function (called center/prototype loss) to improve recognition performance. The class means are then constrained to a sphere by Liu et. al in SphereFace (CVPR 2017) which further improves the performance (like the ETF proposed in this paper). Deng et al. in ArcFace (CVPR 2019) further add an angular penalty between points on a sphere. There have been several papers after this (look for papers citing these 4) which explore ways to do classification without a linear layer followed by softmax+CE which are based on per-class features and their margins.

[1] A Discriminative Feature Learning Approach for Deep Face Recognition, Wen et. al, ECCV 2016

[2] Prototypical networks for few-shot learning, Snell et. al, Neurips 2017

[3] SphereFace: Deep Hypersphere Embedding for Face Recognition, Liu et. al, CVPR 2017

[4] ArcFace: Additive Angular Margin Loss for Deep Face Recognition, Deng et. al, CVPR 2019

---

> ### Author Response · Authors · 2022-08-02
> **Response to your review (Part 1)**
>
> Dear Reviewer,
>
> Thanks for your valuable comments.
>
>
> + The true contributions of our paper.
>
> We notice that our paper only scores 1 on “contribution” in your review. We think that you may misunderstand our work and overlook some important contributions. So, we would like to first re-summarize our contributions, and then respond to your questions carefully.
>
> **Note that the aim of our study is NOT to propose an algorithm for some application task, nor to study the classifier prototype itself.** The following contributions are consistent with the ones claimed in our paper (lines 78 - 90), but highlight the overlooked points in more details.
>
> **[C1]** Neural collapse as an elegant phenomenon is observed by [28], and is proved (within the LPM) to be the global optimality of training on a balanced dataset under the CE loss [5, 8, 22, 15, 47] or MSE loss [24, 37], which theoretically explains why neural collapse happens in balanced training. As far as we know, we are the first to show that neural collapse can even happen in imbalanced training as long as the learnable classifier is fixed as an ETF (Theorem 1 and Remark 1 in our paper, proved in Appendix A).
>
> **[C2]** Our theoretical analyses on the gradient of CE loss indicate that: (1) neural collapse will be broken in imbalanced training with a learnable classifier, i.e., classier vectors of minor classes would be close or even merged, due to the imbalanced gradients with respect to the learnable classifier; in contrast, our fixed ETF classifier does not suffer from this dilemma (Remark 2 in our paper); (2) the emergence of neural collapse in balanced training is attributed to the “pull-push” mechanism in the CE loss (Remark 3 in our paper); (3) when the classifier is fixed as an ETF optimality, the "pull" gradient is always accurate, and the "push" gradient is no longer necessary, which inspires us to develop a new loss function with more accurate gradient.
>
> **[C3]** Inspired by (3) in [C2], we further develop a new loss function with only "pull" gradient and the same optimality as the CE loss. It has a better convergence property than the CE loss, which is theoretically proved (Theorem 2 in our paper, proved in Appendix B).
>
> **[C4]** Experiments of long-tail classification on CIFAR-10, CIFAR-100, SVHN, STL, and ImageNet are conducted to verify our theories and theory-inspired methods in [C1]-[C3].
>
> We know that many variants on classifier prototype and loss function have been proposed for application tasks such as long-tailed classification, few-shot learning, face recognition, and maybe domain adaptation. However, we think our work should not be judged only on what method we use and what performance we achieve. **Our theoretical results including Theorem 1, Remark 1, Remark 2, Remark 3, and Theorem 2, are original, and should not be overlooked.** But we do not receive a feedback on any of them from your review.
>
> We believe that the contributions, in particular [C1]-[C3], are advancements for the neural collapse area because:
>
> (1) Current neural collapse studies only focus on why neural collapse happen in balanced training [5, 8, 22, 15, 47, 24, 37], while we are the first to show that neural collapse can also be a global optimality in imbalanced training;
>
> (2) Compared with current neural collapse studies, we not only show the neural collapse global optimality, but also inspire a new loss function whose benefit is provable;
>
> (3) There are only empirical experiments in these studies showing the convergence to neural collapse, while we additionally show practical applicability of our neural collapse inspired methods by long-tailed experiments on multiple datasets including ImageNet.
>
> **Based on the statement above, we hope that our contributions can be properly recognized.**
>
> ----
> References
>
> [5] C. Fang, H. He, Q. Long, and W. J. Su. Exploring deep neural networks via layer-peeled model: Minority collapse in imbalanced training. Proceedings of the National Academy of Sciences, 118(43), 2021.
>
> [8] F. Graf, C. Hofer, M. Niethammer, and R. Kwitt. Dissecting supervised constrastive learning. In ICML, pages 3821–3830. PMLR, 2021.
>
> [15] W. Ji, Y. Lu, Y. Zhang, Z. Deng, and W. J. Su. An unconstrained layer-peeled perspective on neural collapse. In ICLR, 2022.
>
> [22] J. Lu and S. Steinerberger. Neural collapse with cross-entropy loss. arXiv preprint arXiv:2012.08465, 2020.
>
> [24] D. G. Mixon, H. Parshall, and J. Pi. Neural collapse with unconstrained features. arXiv preprint, arXiv:2011.11619, 2020.
>
> [28] V. Papyan, X. Han, and D. L. Donoho. Prevalence of neural collapse during the terminal phase of deep learning training. Proceedings of the National Academy of Sciences, 2020.
>
> [37] T. Tirer and J. Bruna. Extended unconstrained features model for exploring deep neural collapse. ICML, 2022.
>
> [47] Z. Zhu, T. DING, J. Zhou, X. Li, C. You, J. Sulam, and Q. Qu. A geometric analysis of neural collapse with unconstrained features. In NeurIPS, 2021.

---

> > ### Author Response · Authors · 2022-08-02
> > **Response to your review (Part 2)**
> >
> > + There are several papers that have evaluated similar loss functions not cited and compared with.
> >
> > Thanks for reminding us of these papers. Indeed, many variants of classifier prototype and loss function have been proposed for application tasks such as long-tailed classification, few-shot learning, face recognition, and maybe domain adaptation. We propose the practice of fixing a learnable classifier as an ETF in order to study the neural collapse optimality in imbalanced training. Although these studies are not related to neural collapse, they are also concerned with classifier and loss function. We will cite these papers and discuss the differences in details.
> >
> > Here we compare with the methods in references [A1,A2,A3,A4] mentioned by you and also [A5] mentioned by Reviewer amWi.
> >
> > [A1] is a pioneering work for face recognition. In [A1], the whole loss function is composed of a standard CE loss and a center loss. The center loss is a MSE function that regresses each feature into its corresponding prototype (center). However, the centers are learnable, jointly with the training of backbone and classifier. In contrast, we fix the classifier vectors as an ETF structure, so they are not learnable. The ETF structure corresponds to the largest pair-wise equiangular separation of $K$ vectors in $\\mathbb{R}^d$. Our dot-regression (DR) loss regresses the inner product of feature and the fixed classifier vector, instead of the feature itself as the center loss in [A1]. Our DR loss only has “pull” gradient term, while all CE-based losses have both “pull” and “push” gradients. The benefit of DR loss over CE loss has been theoretically proved (Theorem 2 in our paper). So, our methods are different from [A1]. Our contributions [C1]-[C4] have no overlap with [A1].
> >
> > [A2] is a study for few-shot learning. They train a network on train set, and then extract features on support set. The mean of features in each class is used as the prototype for that class. Each sample in query set is classified by measuring the distance between its feature and each prototype. In contrast, our prototypes are fixed as an ETF, instead of the means of features in each lass. Our proposed DR loss has no similarity to [A2] that uses CE loss for training the backbone and Euclidean distance for classifying query samples. So, our methods are different from [A2]. Our contributions [C1]-[C4] have no overlap with [A2].
> >
> >
> > [A3] and [A4] are studies for face recognition. They normalize all classifier prototypes as 1 into a sphere. However, the classifier is still learnable in [A3] and [A4]. Although the ETF structure has equal $\\ell_2$ norms, which means the prototypes are also located on a sphere, we fix the classifier as an ETF. The classifier in our case is not learnable. We prove that neural collapse can inherently happen even in imbalanced training as long as the classifier is fixed as an ETF (Theorem 1 in our paper). So, our methods are different from [A3] and [A4]. Our contributions [C1]-[C4] have no overlap with [A3] and [A4].
> >
> >
> > [A5] is a study for face recognition. They study the cosine softmax loss where both feature and classifier prototype are normalized. They propose a loss function whose gradients have coefficients of similarity instead of probability as in the original CE loss. Still, the classifier is [A5] is learnable, while we fix the classifier as an ETF to induce the neural collapse optimality in imbalanced training. The only similarity is that the gradients of our proposed DR loss also have coefficients of cosine similarity (shown in Eq. (15) in our paper). However, the loss proposed in [A5] still relies on the “push” gradient term, while DR loss only has “pull” gradient term that always directs to the optimality. The benefit of DR loss has been theoretically proved. So, our methods are different from [A5]. Our contributions [C1]-[C4] have no overlap with [A5].
> >
> > Finally, we would like to highlight that for any classification problem, the goal is to produce discriminative features with within-variance minimized and between-variance maximized, just as described by the neural collapse phenomenon. So, our method that tries to induce neural collapse in imbalanced training will inevitably share similar spirit with those studies in application areas such as few-shot learning and face recognition. **However,** as far as we know, our method of fixing a classifier as an ETF has only been adopted in [47] (as mentioned in our Related Work section, Lines 111-115). But [47] only tries this practice in experiment to show no harm of performance, while we show its ability to induce neural collapse optimality in imbalanced training. Besides, our method of the DR loss has not been proposed in any prior study. **Therefore, our methods (ETF classifier + DR loss) and our contributions ([C1]-[C4]) should be properly recognized.**
> >
> > -----
> > References
> >
> > [A1, A2, A3, A4, A5] are listed in the next part of response.

---

> > > ### Author Response · Authors · 2022-08-02
> > > **Response to your review (Part 3)**
> > >
> > > + It is not clear why would features collapse to within-class mean? What is the loss function being used to do so? Are class labels even used during training which leads to this equiangular tight frame? The reader needs to be given a high level introduction to ETF.
> > >
> > >
> > > Please note that we do give the detailed backgrounds of neural collapse, equiangular tight frame (ETF), and layer-peeled model (LPM) in Section 3 in our paper. It covers nearly one page from Line 116 to Line 158.
> > >
> > > Neural collapse as an elegant phenomenon is observed by [28]. But [28] does not give the answer to why neural collapse happen (why would features collapse to within-class mean). In later studies, using the LPM analytical tool, neural collapse is proved to be the global optimality of training on a balanced dataset under the CE loss [5, 8, 22, 15, 47] or MSE loss [24, 37], which theoretically explains why neural collapse happens in balanced training (accordingly also explains why would features collapse to within-class mean). So, both CE and MSE losses can induce neural collapse in balanced training, and labels are necessary. The motivation of our study is to induce neural collapse in imbalanced training. As claimed in [C1], as far as we know, we are the first to show that neural collapse can even happen in imbalanced training as long as the learnable classifier is fixed as an ETF
> > >
> > >
> > > ----
> > >
> > > References
> > >
> > > [A1] A Discriminative Feature Learning Approach for Deep Face Recognition, Wen et. al, ECCV 2016
> > >
> > > [A2] Prototypical networks for few-shot learning, Snell et. al, Neurips 2017
> > >
> > > [A3] SphereFace: Deep Hypersphere Embedding for Face Recognition, Liu et. al, CVPR 2017
> > >
> > > [A4] ArcFace: Additive Angular Margin Loss for Deep Face Recognition, Deng et. al, CVPR 2019
> > >
> > > [A5] P2SGrad: Refined Gradients for Optimizing Deep Face Models, Xiao Zhang et al., CVPR 2019
> > >
> > > [5] C. Fang, H. He, Q. Long, and W. J. Su. Exploring deep neural networks via layer-peeled model: Minority collapse in imbalanced training. Proceedings of the National Academy of Sciences, 118(43), 2021.
> > >
> > > [8] F. Graf, C. Hofer, M. Niethammer, and R. Kwitt. Dissecting supervised constrastive learning. In ICML, pages 3821–3830. PMLR, 2021.
> > >
> > > [15] W. Ji, Y. Lu, Y. Zhang, Z. Deng, and W. J. Su. An unconstrained layer-peeled perspective on neural collapse. In ICLR, 2022.
> > >
> > > [22] J. Lu and S. Steinerberger. Neural collapse with cross-entropy loss. arXiv preprint arXiv:2012.08465, 2020.
> > >
> > > [24] D. G. Mixon, H. Parshall, and J. Pi. Neural collapse with unconstrained features. arXiv preprint, arXiv:2011.11619, 2020.
> > >
> > > [28] V. Papyan, X. Han, and D. L. Donoho. Prevalence of neural collapse during the terminal phase of deep learning training. Proceedings of the National Academy of Sciences, 117(40):24652–24663, 2020.
> > >
> > > [37] T. Tirer and J. Bruna. Extended unconstrained features model for exploring deep neural collapse. ICML, 2022.
> > >
> > > [47] Z. Zhu, T. DING, J. Zhou, X. Li, C. You, J. Sulam, and Q. Qu. A geometric analysis of neural collapse with unconstrained features. In NeurIPS, 2021.

---

> > > ### Comment · Reviewer_MaCE · 2022-08-08
> > > **.**
> > >
> > > [A3] and [A4] are studies for face recognition. They normalize all classifier prototypes as 1 into a sphere. However, the classifier is still learnable in [A3] and [A4]. Although the ETF structure has equal norms, which means the prototypes are also located on a sphere, we fix the classifier as an ETF. The classifier in our case is not learnable. We prove that neural collapse can inherently happen even in imbalanced training as long as the classifier is fixed as an ETF (Theorem 1 in our paper). So, our methods are different from [A3] and [A4].
> > >
> > >
> > >
> > > From what I am reading, the conclusion of this paper is that not providing flexibility to the vectors in angular loss functions in A3, A4, A5 by keeping them fixed as an ETF is better than them being learnable. If having them learnable is not advantageous, in my opinion it would be good to compare against these loss functions on some of the benchmarks where these methods were tested instead of comparing with CE loss on Imagenet.
> > >
> > > A3, A4, A5 are clearly the closest competitors of this ETF based method instead of the CE loss, but these methods have not even been cited, let alone having an experimental comparison.

---

> > > > ### Author Response · Authors · 2022-08-08
> > > > **Comparison with the face recognition studies**
> > > >
> > > > Sorry for the late results. It takes us some days to re-implement the face recognition methods. We compare our method with these face recognition methods on long-tailed classification, general classification, and face recognition. The results are shown as follows:
> > > >
> > > > (1)	Long-tailed classification
> > > >
> > > > The experiments of long-tailed classification are conducted on CIFAR-10 with different imbalance ratios. All models are trained with ResNet-32 backbone under the same setting.
> > > >
> > > > | Imbalance ratio | 0.005 | 0.01 | 0.02 |
> > > > | --- | --- | --- | --- |
> > > > |Learnable Classifier + Weighted CE Loss | 68.5$\\pm$0.3 | 73.9$\\pm$0.3 |79.3$\\pm$0.2|
> > > > |Learnable Classifier + Focal Loss [A7] | 69.9$\\pm$0.2 | 75.4$\\pm$0.3 | 78.9$\\pm$0.2 |
> > > > |Learnable Classifier + CB Loss [A6] | 69.3$\\pm$0.4 | 76.0$\\pm$0.2 | 79.5$\\pm$0.4 |
> > > > | Center Loss [A1] | 66.8$\\pm$0.5 | 70.6$\\pm$0.2 | 77.9$\\pm$0.2 |
> > > > | SphereFace Loss [A3] | 67.4$\\pm$0.3 | 73.0$\\pm$0.3 | 78.9$\\pm$0.4|
> > > > | ArcFace Loss [A4] | 66.3$\\pm$0.6 | 71.9$\\pm$0.4 | 77.4$\\pm$0.4|
> > > > | P2SGrad [A5] |  66.8$\\pm$0.3 | 69.8$\\pm$0.3 | 76.6$\\pm$0.3|
> > > > |ETF Classifier + DR Loss (ours) | 71.9$\\pm$0.3 | 76.5$\\pm$0.3 | 81.0$\\pm$0.2 |
> > > >
> > > > (2)	General classification
> > > >
> > > > We perform the general classification using ResNet-50 on ImageNet. The results are shown as follows:
> > > >
> > > > |Methods| Top-1 Accuracy |
> > > > |---|---|
> > > > | Learnable Classifier + CE Loss | 76.21 |
> > > > | Center Loss [A1] | 75.46 |
> > > > | SphereFace Loss [A3] | 75.71 |
> > > > | ArcFace Loss [A4] | 75.70 |
> > > > | P2SGrad [A5] | 75.66 |
> > > > | ETF classifier + DR Loss (ours) | 76.05 |
> > > >
> > > > (3)	Face recognition
> > > >
> > > > We use the code base released by ArcFace to conduct experiments of face recognition on LFW and YTF. We train on MS1MV2 and adopt the training setting used in ArcFace. The results are shown as follows:
> > > >
> > > > | Methods | LFW | YTF |
> > > > |--- | --- | --- |
> > > > | Center Loss [A1] | 99.28 | 94.9 |
> > > > | SphereFace [A3] | 99.42 | 95.0 |
> > > > | ArcFace [A4] | 99.83 | 98.0 |
> > > > |P2SGrad [A5] | 99.82 | 97.2 |
> > > > |ETF classifier + DR Loss (ours) | 99.82 | 97.7 |
> > > >
> > > > It is shown that our method achieves the best in long-tailed classification. In general learning on ImageNet, our method is close to the baseline (Learnable classifier + CE loss), while the face recognition methods have degraded performances than the baseline. In face recognition, it is shown that our method is still competitive with the losses proposed in [A1, A3, A4, A5]. We believe that the performance of our method can be further improved if more engineering work, such as carefully tunning the learning rate, is involved.
> > > >
> > > > Please note that we **never claim** that “learnable classifier is not advantageous in all cases”. Our contributions have been rigorously stated in C1 – C4 and Lines 78-90 in our paper. We prove that fixing an ETF classifier is advantageous because neural collapse can inherently happen even in **imbalanced training.** So, we mainly focused on long-tailed classification experiments and did not test it on face recognition in our paper.
> > > >
> > > > Our method also works well in general learning and face recognition because it better converges to the neural collapse optimality. As described by neural collapse, the ETF structure corresponds to the maximal pair-wise equiangular separation. It is consistent with the goal of face recognition that collapsing features of each class into a center, while maximizing the distances between centers of different classes. Our method of fixing an ETF classifier directly allocates an optimality and then learns features towards it. Besides, our DR loss has an advantage in convergence over the CE loss (Theorem 2). As stated in Sec4.3, DR loss only has "pull" gradient, while the CE loss and all the variants in the face recognition studies [A1, A3, A4, A5] rely on the "push" gradient whose direction may be inaccurate.
> > > >
> > > > So, we think applying our method into face recognition is attractive as a future study. **We will cite these studies and include the results above in our revised paper.**
> > > >
> > > > ---
> > > > Reference
> > > >
> > > > [A1] A Discriminative Feature Learning Approach for Deep Face Recognition, Wen et. al, ECCV 2016
> > > >
> > > > [A3] SphereFace: Deep Hypersphere Embedding for Face Recognition, Liu et. al, CVPR 2017
> > > >
> > > > [A4] ArcFace: Additive Angular Margin Loss for Deep Face Recognition, Deng et. al, CVPR 2019
> > > >
> > > > [A5] P2SGrad: Refined Gradients for Optimizing Deep Face Models Xiao Zhang, Rui Zhao, Junjie Yan, Mengya Gao, Yu Qiao, Xiaogang Wang, Hongsheng Li; Proceedings of the IEEE/CVF Conference on Computer Vision and Pattern Recognition (CVPR), 2019
> > > >
> > > > [A6] Class-Balanced Loss Based on Effective Number of Samples, Cui et al., CVPR 2019.
> > > >
> > > > [A7] Focal loss for dense object detection, Lin et al., ICCV 2017.

---

> > > > > ### Comment · Reviewer_MaCE · 2022-08-08
> > > > > **Evaluations**
> > > > >
> > > > > LFW is too small to compare as most of these methods would give similar results. A3/A4 are similar on YTF.
> > > > >
> > > > > Megaface / IJB is where these methods are typically compared where differences are more  obvious across methods, Table 6 https://openaccess.thecvf.com/content_CVPR_2019/papers/Deng_ArcFace_Additive_Angular_Margin_Loss_for_Deep_Face_Recognition_CVPR_2019_paper.pdf

---

> > > > > > ### Author Response · Authors · 2022-08-09
> > > > > > **Response to your concern on large-scale face recognition experiment**
> > > > > >
> > > > > > Results on MegaFace is shown as follows:
> > > > > >
> > > > > > | Methods | Id(\%) | Ver (\%) |
> > > > > > |---|---|---|
> > > > > > |Center Loss |65.49 | 80.14 |
> > > > > > | Sphere Loss | 72.73 | 85.56 |
> > > > > > |ArcFace Loss | 81.03 | 96.98 |
> > > > > > | Our Method | 80.91 | 96.98 |
> > > > > >
> > > > > > It is shown that our method is also applicable to large-scale face recognition.
> > > > > >
> > > > > > We do not think our contributions can be denied just because we do not offer SOTA performance on face recognition. **As stated in our previous responses, our study has no relation to face recognition. We did not claim any contribution related to face recognition. Our method is DIFFERENT from the loss variants in current face recognition studies.** Therefore, whether we surpass the face recognition methods on even challenging large face recognition datasets makes no difference to the contributions of our study.
> > > > > >
> > > > > > **We have repeated our contributions [C1]-[C4] for multiple times. But all your comments are only concerned with face recognition. We are obliged to say that our main contributions are totally overlooked.**

---

> > > > > > > ### Comment · Reviewer_MaCE · 2022-08-09
> > > > > > > **Our study has no relation to face recognition**
> > > > > > >
> > > > > > > Changing labels to face identities would affect the loss function adversely but somehow it would work better if it is internet tags? Claiming that internet image tags are better suited for this loss function but it somehow does not work for face identities seems like something is not right. If the method really generalizes (for example adding residual connections in CNNs, or using a deeper network vs shallower, less vs more training data), it would not matter if the class labels are human identities or internet tags and we would observe an improvement in performance regardless of the application, especially for something as generic of a module like a loss function! Probably this is just an over tuned method on a couple of datasets (that too on weak baselines even for those datasets) claiming to be a better loss function, when in reality it is not.

---

> > > > > > > > ### Author Response · Authors · 2022-08-09
> > > > > > > > **A factual error in your comment**
> > > > > > > >
> > > > > > > > As stated in your comment, a better loss function would have an improvement in performance regardless of the application task, no matter if the labels are human or tags. We would like to point out that it is a factual error. As a widely adopted practice, people use Focol loss for long-tailed classification, but will not use it for general classification or face recognition. People use ArcFace for face recognition, but will not use it for long-tailed classification or general classification. If your comment were right, ArcFace would have the best performance no matter on face recognition or long-tailed classification. But actually, as shown in the long-tailed experiments in our previous response, our method performs much better than the weighted CE baseline (71.9 vs 68.5 in 0.005 imbalance ratio), while ArcFace is apprantly worse than the weighted CE baseline (66.3 vs 68.5).
> > > > > > > >
> > > > > > > > So, your comment that a loss will be better or worse regardless of the application task and your criticism that our method also needs to be better on face recognition are groundless.

---

> > > > > > > > > ### Comment · Reviewer_MaCE · 2022-08-09
> > > > > > > > > **groundless comments**
> > > > > > > > >
> > > > > > > > > All those loss functions target an application like long tail classification, face recognition and show meaningful improvements on well recognized benchmarks for those applications. If the claim here is that we are improving image classification instead of a loss function which makes generic improvements, I would expect improving state of the art for image classification, otherwise if its a generic loss function which works everywhere, then improvements should be applicable across several applications. But this method is neither generic nor does it improve an application (like image classification).

---

> > > > > > > > > > ### Author Response · Authors · 2022-08-09
> > > > > > > > > > **Not all studies must target an application and improve its SOTA performance**
> > > > > > > > > >
> > > > > > > > > > We have reminded for multiple times of our contributions [C1]-[C4]. We never claim that we want to improve an application. Not all studies must target an application and improve its SOTA performance. If our contributions [C1]-[C4] are properly recognized, you will find that **our work is not an application study at all.**

---

> > > > > > > > > > > ### Comment · Reviewer_MaCE · 2022-08-09
> > > > > > > > > > > **contributions**
> > > > > > > > > > >
> > > > > > > > > > > But the empirical data to back those claims is conflicting. The claims work on one dataset but break on another. Moreover, compared to methods which are the closest in theory (I would say methods like ArcFace are more similar to ETF than CE loss), the method shows no improvement, breaking the theory.

---

> > > > > > > > > > > > ### Author Response · Authors · 2022-08-09
> > > > > > > > > > > > **Your comment that a loss will be better or worse regardless of the application task and your criticism that our method also needs to be the best on face recognition are groundless.**
> > > > > > > > > > > >
> > > > > > > > > > > > Our method performs much better than the weighted CE baseline (71.9 vs 68.5 in 0.005 imbalance ratio), while ArcFace is apprantly worse than the weighted CE baseline (66.3 vs 68.5). Is this a confilct for ArcFace?
> > > > > > > > > > > >
> > > > > > > > > > > > Our Theorem 1 mainly focuses on the imbalanced training setting, compared with previous studies on neural collapse that can only deal with balanced training. Requiring our method to be the best on face recognition is as groundless as requiring ArcFace to work in long-tailed classification. Besides, in the three cases (long-tailed classification, general classification, face recognition), our method is competitive in all three cases, and surpasses the face recognition methods in both long-tail classification and general classification. As a comparison, face recognition methods like ArcFace can only work in face recognition. They perform significantly worse in long-tailed classification or general classification.
> > > > > > > > > > > >
> > > > > > > > > > > > So, a loss cannot be always the best regardless of the application task. Requiring our method also to be the best on face recognition is groundless. A theory can be broken only when its proof is wrong.

---

### Official Review · Reviewer_amWi · 2022-07-11

**Rating:** 6
**Confidence:** 5
**Soundness:** 3 good
**Presentation:** 3 good
**Contribution:** 2 fair

**Summary:**

The paper studies neural collapse in the case in which the final classifier is not subject to back-propagation and is initialized with the vertices of a simplex equiangular tight frame (ETF). In particular the paper studies the unbalanced learning regime and shows that the neural collapse properties hold also in this case. The proof is based on the layered-peeled model assumption.


**Questions:**

See above.

**Limitations:**

No noteworthy limitations or potential negative societal impacts.

**Strengths And Weaknesses:**

STRENGTHS:

1) The paper is very interesting as it provides a theoretical analysis of the class imbalance problem with a fixed classifier inspired by the recent neural collapse phenomenon.

2) The push-pull mechanism from the gradient analysis is novel and it is a clear contribution of the paper.

3) The idea may also have practical implications related to some recent works about learning compatible features representations [C, D]. Compatible features are learned so that they can be directly compared even if they are learned from different time instants or from different network architectures. The method [D] specifically takes advantage of a fixed classifier exactly defined as the one proposed in [47] and in this submission.

WEAKNESSES:

1) The paper contributions should be improved. It seems that the main contribution of the paper is about the combination of the following three aspects: neural collapse, unbalanced data and fixed classifiers. The title does not convey a direct relationship with these three aspects. In its current shape it seems more referring to the fixed classifier aspect, like in [29] and [12], rather than to the imbalanced Neural Collapse training regime. This is mostly due to the fact that if a learnable classifier is not “really needed” it is supposed that a fixed one would be preferred. In the reviewer's opinion the title should be improved reflecting more the main content of the paper.
In this respect, the differences with [47] and [29] should be better highlighted. In particular a question that should be answered is: what is the difference between an ETF simplex and the classic d-simplex regular polytope? The question is valid either in the fixed case and in the learnable one. Finally, as the paper [5] is very related with the imbalance aspect of the paper, it should be discussed in the related work section and possibly differences highlighted.

2) It seems that in many papers (including the seminal one [28] and [47]) the relationship between the number of classes K and the dimension d required to observe the neural Collapse phenomenon is d≥K while in the submitted manuscript and in [47] d≥K-1. Is there any specific motivation for these two cases?

3) Some important references are missing: [A,B]. Although not directly related to neural collapse, the paper [A] proposes a similar gradient analysis of the final classifier. A brief discussion highlighting differences and contributions should be given. The paper [B] seems to be one of the first published papers on maximal separation and fixed classifiers with the simplex geometry.

4) Is the random ETF initialization needed to avoid possible biases in its choice? Is there any evaluation quantifying this dependency?

5) Eq. 15 is referring to the optimization of the angle between features and classifier prototypes. What is the difference between that loss and the classic entropy loss in which both features and prototypes are normalized? Both minimize the angles between features and prototypes.

References

[A] P2SGrad: Refined Gradients for Optimizing Deep Face Models Xiao Zhang, Rui Zhao, Junjie Yan, Mengya Gao, Yu Qiao, Xiaogang Wang, Hongsheng Li; Proceedings of the IEEE/CVF Conference on Computer Vision and Pattern Recognition (CVPR), 2019

[B] Maximally Compact and Separated Features with Regular Polytope Networks Federico Pernici, Matteo Bruni, Claudio Baecchi, Alberto Del Bimbo; Proceedings of the IEEE/CVF Conference on Computer Vision and Pattern Recognition (CVPR) Workshops, 2019, pp. 46-53

[C] Shen Y, Xiong Y, Xia W, Soatto S. Towards backward-compatible representation learning. InProceedings of the IEEE/CVF Conference on Computer Vision and Pattern Recognition 2020 (pp. 6368-6377).

[D] Biondi N, Pernici F, Bruni M, Del Bimbo A. CoReS: Compatible Representations via Stationarity. arXiv preprint arXiv:2111.07632. 2021 Nov 15.

---

> ### Author Response · Authors · 2022-08-02
> **Response to your review (Part 1)**
>
> Dear reviewer,
>
> Thanks for your valuable comments.
>
> + The contributions of this paper.
>
> We deeply thank you for appreciating our contribution of the gradient analysis on the "pull-push" mechanism. For your concern about the contributions, we would like to re-summarize our contributions. The following contributions are consistent with the ones claimed in our paper (lines 78 - 90), but highlight the overlooked points in more details.
>
> **[C1]** Neural collapse as an elegant phenomenon is observed by [28], and is proved (within the LPM) to be the global optimality of training on a balanced dataset under the CE loss [5, 8, 22, 15, 47] or MSE loss [24, 37], which theoretically explains why neural collapse happens in balanced training. As far as we know, we are the first to show that neural collapse can even happen in imbalanced training as long as the learnable classifier is fixed as an ETF (Theorem 1 and Remark 1 in our paper, proved in Appendix A).
>
> **[C2]** Our theoretical analyses on the gradient of CE loss indicate that: (1) neural collapse will be broken in imbalanced training with a learnable classifier, i.e., classier vectors of minor classes would be close or even merged, due to the imbalanced gradients with respect to the learnable classifier; in contrast, our fixed ETF classifier does not suffer from this dilemma (Remark 2 in our paper); (2) the emergence of neural collapse in balanced training is attributed to the “pull-push” mechanism in the CE loss (Remark 3 in our paper); (3) when the classifier is fixed as an ETF optimality, the "pull" gradient is always accurate, and the ``push’’ gradient is no longer necessary, which inspires us to develop a new loss function with more accurate gradient.
>
> **[C3]** Inspired by (3) in [C2], we further develop a new loss function with only "pull" gradient and the same optimality as the CE loss. It has a better convergence property than the CE loss, which is theoretically proved (Theorem 2 in our paper, proved in Appendix B).
>
> **[C4]** Experiments of long-tail classification on CIFAR-10, CIFAR-100, SVHN, STL, and ImageNet are conducted to verify our theories and theory-inspired methods in [C1]-[C3].
>
> We believe that the contributions [C1]-[C4] are advancements for the neural collapse area.
>
> + The current title more refers to the fixed classifier aspect, rather than to the imbalanced neural collapse training regime.
>
> Yes, we really thank you for the suggestion on our title. As responded in the previous question, our contributions are mainly targeted at the neural collapse area. **The aim of our study is NOT to propose an algorithm for some application task, nor to study the fixed classifier itself.** We think that the contributions should be conveyed in the Introduction section (lines 78 - 90), instead of the title, so we did not make a title carefully. We will follow your suggestion and revise it accordingly. A title such as “neural collapse in imbalanced training” is more preferable.
>
> ----
> References
>
> [5] C. Fang, H. He, Q. Long, and W. J. Su. Exploring deep neural networks via layer-peeled model: Minority collapse in imbalanced training. Proceedings of the National Academy of Sciences, 118(43), 2021.
>
> [8] F. Graf, C. Hofer, M. Niethammer, and R. Kwitt. Dissecting supervised constrastive learning. In ICML, pages 3821–3830. PMLR, 2021.
>
> [15] W. Ji, Y. Lu, Y. Zhang, Z. Deng, and W. J. Su. An unconstrained layer-peeled perspective on neural collapse. In ICLR, 2022.
>
> [22] J. Lu and S. Steinerberger. Neural collapse with cross-entropy loss. arXiv preprint arXiv:2012.08465, 2020.
>
> [24] D. G. Mixon, H. Parshall, and J. Pi. Neural collapse with unconstrained features. arXiv preprint, arXiv:2011.11619, 2020.
>
> [28] V. Papyan, X. Han, and D. L. Donoho. Prevalence of neural collapse during the terminal phase of deep learning training. Proceedings of the National Academy of Sciences, 117(40):24652–24663, 2020.
>
> [29] F. Pernici, M. Bruni, C. Baecchi, and A. Del Bimbo. Regular polytope networks. IEEE Transactions on Neural Networks and Learning Systems, 2021
>
> [37] T. Tirer and J. Bruna. Extended unconstrained features model for exploring deep neural collapse. ICML, 2022.
>
> [47] Z. Zhu, T. DING, J. Zhou, X. Li, C. You, J. Sulam, and Q. Qu. A geometric analysis of neural collapse with unconstrained features. In NeurIPS, 2021.

---

> > ### Author Response · Authors · 2022-08-02
> > **Response to your review (Part 2)**
> >
> > + The difference with [47] and [29] should be better highlighted. What is the difference between an ETF simplex and the classic d-simplex regular polytope?
> >
> > The objective of LPM studied in [47] is the CE loss with the regularizations of features and classifier. They prove that: (1) neural collapse is the global optimality of the objective in the LPM when training on a balanced dataset; (2) despite being nonconvex, the landscape of the objective in this case is benign, which means that there is no spurious local minimum, so gradient-based optimization method can easily escape from the strict saddle points to look for the global minimizer.
> >
> > In contrast, the objective of LPM we consider is the CE loss with the constraints of features and classifier. We mainly consider neural collapse in the imbalanced training. Our contributions are summarized in [C1]-[C4], which are different from the two results in [47].
> >
> > [29] shows that fixing a learnable classifier as the vertices of regular polytopes, including d-simplex, d-cube, and d-orthoplex, helps to learn stationary and maximally separated features. It does not harm the performance, and in many cases improves the performance. It also brings faster speed of convergence and reduces the model parameters. Their spirit of learning maximally separated features is very similar to neural collapse. Compared with [29], we prove that neural collapse can even be the global optimality of the CE loss in the imbalanced training using the LPM analytical tool. We also propose a new loss function with a provable advantage over the CE loss.
> >
> > In neural collapse studies, the simplex ETF is a structure where $K$ vertices in $\\mathbb{R}^d$ have the same length and have the largest pair-wise equiangular separation. In [29], d-simplex is a generalization of the triangle or tetrahedron definition to the dimension in $\\mathbb{R}^d$, such as a triangle in  $\\mathbb{R}^2$ and a tetrahedron in $\\mathbb{R}^3$. For ETF, when $d=K-1$, the ETF reduces to a regular simplex. For example, when $d=3$ and $K=4$, the ETF is a just a tetrahedron, which is the same as 3-simplex. But ETF allows the condition that there are a less number of vertices than a $d$-simplex, e.g., $d=3$ and $K=3$. As long as $d\\ge K-1$, we can have an ETF according to Eq. (1) in our paper. But a $d$-simplex only holds when $d=K-1$, which may limit the choice of dimension for a classification problem with a given number of classes.
> >
> > + Highlight the difference with [5].
> >
> > The objective of LPM studied in [5] is also the CE loss with feature and classifier constraints. They prove that (1) neural collapse is the global optimality of the objective when training on a balanced dataset; (2) in imbalanced training, neural collapse will be broken, and the prototypes of minor classes will be merged, which explains the difficulty of imbalanced training.
> >
> > We also study the objective of CE loss with feature and classifier constraints. As a comparison, (1) We prove that neural collapse can also be the global optimality for imbalanced training as long as the classifier is fixed as an ETF (Theorem 1); (2) We analyze from the gradient perspective and show that the broken neural collapse in imbalanced training is caused by the imbalanced magnitude of gradients of the CE loss (Remark2). We also show that the “pull-push” mechanism is crucial for the emergence of neural collapse in the CE loss in balanced training (Remark 3); (3) Inspired by the analyses, we propose a new loss function with a provable advantage over the CE loss (Theorem 2).
> >
> > We will make the comparison with [47, 29, 5] more detailed and highlighted in our revised version.
> >
> > ----
> > References
> >
> > [5] C. Fang, H. He, Q. Long, and W. J. Su. Exploring deep neural networks via layer-peeled model: Minority collapse in imbalanced training. Proceedings of the National Academy of Sciences, 118(43), 2021.
> >
> > [29] F. Pernici, M. Bruni, C. Baecchi, and A. Del Bimbo. Regular polytope networks. IEEE Transactions on Neural Networks and Learning Systems, 2021
> >
> > [47] Z. Zhu, T. DING, J. Zhou, X. Li, C. You, J. Sulam, and Q. Qu. A geometric analysis of neural collapse with unconstrained features. In NeurIPS, 2021.

---

> > > ### Author Response · Authors · 2022-08-02
> > > **Response to your review (Part 3)**
> > >
> > > + Is there any specific motivation for the two cases, $d\\ge K$ and $d\\ge K-1$
> > >
> > > $d\\ge K-1$ is a necessary condition for $K$ vectors in $\\mathbb{R}^d$ to form an ETF structure that satisfies Eq. (2) in our paper. It is the most widely adopted case in neural collapse studies. In [28], a more strict case $d\\ge K$ is used to allow the rotation by the partial orthogonal matrix $\\mathbf{U}$ such that $\\mathbf{U}^T\\mathbf{U}=\\mathbf{I}$.
> > >
> > > + Important references are missing [A,B]
> > >
> > > Thanks for reminding us of these works. We will cite the two references and compare our contributions with theirs in details.
> > >
> > > In the cosine softmax loss, the logit is the cosine between feature and classifier prototypes multiplied by a hyperparameter. In the CE loss, the logit is inner product of feature and classifier prototypes.
> > >
> > > [A] analyzes the gradients of the cosine softmax loss and proposes a new loss function whose gradients have coefficients of similarity instead of probability as in the original CE loss.
> > >
> > > As a comparison, we conduct theoretical analyses on the gradient of CE loss, instead of the cosine softmax loss in [A]. The gradients of our proposed loss, DR loss in Eq. (15), also have coefficients of cosine similarity. But compared with [A], DR loss only has “pull” gradient with classifier fixed as an optimality, while the loss proposed in [A] still relies on the “push” gradient term. Besides, our loss is proved to have a better convergence property, while [A] does not give a rigorous support.
> > >
> > > As the earlier work than [29], [B] first shows that fixing a learnable classifier as the vertices of regular polytopes, including d-simplex, d-cube, and d-orthoplex, helps to learn stationary and maximally separated features. The differences have been discussed in the previous response to the comparison with [29]. We cited and compared with their extended work [29], but missed [B]. We will cite [B] and highlight the differences in the revised paper.
> > >
> > >
> > >
> > > + Is the random ETF initialization needed to avoid possible biases in its choice? It there any evaluation quantifying this dependency?
> > >
> > > There is no need to worry about the bias of a random ETF. What matters is the separation structure of the classifier prototypes, instead of their specific directions. As shown in Table 1 in our paper, we run the same model for multiple times. For each run, a random ETF is produced. The standard deviation is low, which means the performance has a low dependence on the ETF with a specific bias.
> > >
> > >
> > > + What is the difference between the loss in Eq. (15) and the classic entropy loss in which both features and prototypes are normalized.
> > >
> > > When feature and prototypes are normalized, the original entropy loss (CE loss) just corresponds to the cosine softmax loss studied in [A]. In this case, the loss still has both “pull” and “push” gradient terms, as shown in the first line of Eq. (3) in paper [A].
> > >
> > > As a comparison, our dot-regression (DR) loss only has the “pull” gradient term and no longer relies on the “push” gradient. The motivation has been stated in the contribution [C2]. When the classifier is fixed as an ETF optimality, the “pull” gradient is always accurate towards the optimality, while “push” gradient does not necessarily direct to the optimality. The advantage of DR loss over the CE loss has been theoretically proved in Theorem 2 in our paper.
> > >
> > > ----
> > > References
> > >
> > > [28] V. Papyan, X. Han, and D. L. Donoho. Prevalence of neural collapse during the terminal phase of deep learning training. Proceedings of the National Academy of Sciences, 117(40):24652–24663, 2020.
> > >
> > > [29] F. Pernici, M. Bruni, C. Baecchi, and A. Del Bimbo. Regular polytope networks. IEEE Transactions on Neural Networks and Learning Systems, 2021
> > >
> > > [A] P2SGrad: Refined Gradients for Optimizing Deep Face Models Xiao Zhang, Rui Zhao, Junjie Yan, Mengya Gao, Yu Qiao, Xiaogang Wang, Hongsheng Li; Proceedings of the IEEE/CVF Conference on Computer Vision and Pattern Recognition (CVPR), 2019
> > >
> > > [B] Maximally Compact and Separated Features with Regular Polytope Networks Federico Pernici, Matteo Bruni, Claudio Baecchi, Alberto Del Bimbo; Proceedings of the IEEE/CVF Conference on Computer Vision and Pattern Recognition (CVPR) Workshops, 2019, pp. 46-53

---

### Official Review · Reviewer_Nj4c · 2022-07-11

**Rating:** 4
**Confidence:** 4
**Soundness:** 2 fair
**Presentation:** 3 good
**Contribution:** 2 fair

**Summary:**

On balanced datasets and when using cross-entropy loss for learning, it is observed that [28] the last layer features from the same class converge to their within-class means and the weights the network in the last layer collapses to the vertices of an equiangular tight frame (ETF). Thus, a natural question to ask is: if this empirical observation can be used to train better deep learning models? The current paper uses this insight to design a network where instead of training for a cross-entropy loss, a loss that regresses the deep features against  a randomly sampled set of vectors (corresponding to the vertices of an ETF) is used for classification. In situations when the data is imbalanced, the paper claims that this prior knowledge of ETF allows better learning of the model. Experiments are provided on small scale datasets (such as CIFAR, STL, etc.) and demonstrate promise of the method in imbalanced settings.

**Questions:**

Please see above.

**Limitations:**

The paper does not describe any limitations as such. However, I believe experiments on larger datasets such as ImageNet could bring out the limitations of the approach better.

**Strengths And Weaknesses:**

I think the insights underlying the paper are encouraging and the application of the ETF model for imbalanced datasets is reasonable. The experiments show improvements on a variety of datasets under imbalanced settings.

While, the insights are interesting, I do not think the contributions are compelling for the following reasons.

1.  The key question the paper asks is L58: "Do we really need to learn a linear classifier at the end of a deep neural network for classification?". While, in the context of ETF, it is an interesting question, however when thinking of this question in the context of learning in general, it is obviously not needed to learn a linear classifier at the end of a DNN. The one-hot encoding typically used on labels in cross-entropy loss based training is one way of encoding with its own geometric meaning, while the one proposed using ETF in the paper is yet another encoding. In that sense, I do not really see a significant contribution the paper brings in. Of course, when using an LPM model, we may get rid of learning of the classifier weights, but when using the method in an end-to-end learning framework, doesn't one still need to learn a method that maps the features into one of the vertices of the ETF?

2.  While ETF is a structure that evolves from the use of CE loss, why not consider potentially other structures if we do not assume any CE loss? Why not a regression into the columns of an arbitrary orthogonal matrix (assuming there is no need to interpret the classification as probabilities any more, and thus there is no need to use a softmax)?

3. There are standard approaches for dealing with the problem of imbalanced datasets, such as choosing weights during training, focal loss, etc. The paper appears to have not considered or compared the proposed approach to any of such alternatives and thus limits the understanding of the benefits to a narrow scope (e.g., Figure 3, Tables 2,3,4, etc.).

4. More experiments, for example in settings when there are very large number of classes as in ImageNet, can make the analysis significantly better. I wonder if in such settings a cross-entropy loss provides better "push" and "pull" gradients (due to the log term) than the proposed linear alternative (14).

Other comments:
1. For the layer-peeled model in (4), it is not clear how precisely are the features produced? If you optimize over W and H, then how do you associate the features to be from the data x? When optimizing for h, are you optimizing the parameters of the underlying network? If so, rewriting h to h_\theta(x) and optimizing over \theta would make the math better. If it is being optimized only on W and H, then how is H conditioned on the data X, and how do you prevent H from being arbitrary?

---

> ### Author Response · Authors · 2022-08-02
> **Response to your review (Part 1)**
>
> Dear Reviewer,
>
> Thanks for your valuable comments.
>
> We think that you may misunderstand our work and overlook some important contributions. So, we would like to first re-summarize our contributions, and then respond to your questions carefully. The following contributions are consistent with the ones claimed in our paper (lines 78 - 90), but highlight the overlooked points in more details.
>
> **[C1]** Neural collapse as an elegant phenomenon is observed by [28], and is proved (within the LPM) to be the global optimality of training on a balanced dataset under the CE loss [5, 8, 22, 15, 47] or MSE loss [24, 37], which theoretically explains why neural collapse happens in balanced training. As far as we know, we are the first to show that neural collapse can even happen in imbalanced training as long as the learnable classifier is fixed as an ETF (Theorem 1 and Remark 1 in our paper, proved in Appendix A).
>
> **[C2]** Our theoretical analyses on the gradient of CE loss indicate that: (1) neural collapse will be broken in imbalanced training with a learnable classifier, i.e., classier vectors of minor classes would be close or even merged, due to the imbalanced gradients with respect to the learnable classifier; in contrast, our fixed ETF classifier does not suffer from this dilemma (Remark 2 in our paper); (2) the emergence of neural collapse in balanced training is attributed to the “pull-push” mechanism in the CE loss (Remark 3 in our paper); (3) when the classifier is fixed as an ETF optimality, the "pull" gradient is always accurate, and the "push" gradient is no longer necessary, which inspires us to develop a new loss function with more accurate gradient.
>
> **[C3]** Inspired by (3) in [C2], we further develop a new loss function with only ``pull’’ gradient and the same optimality as the CE loss. It has a better convergence property than the CE loss, which is theoretically proved (Theorem 2 in our paper, proved in Appendix B).
>
> **[C4]** Experiments of long-tail classification on CIFAR-10, CIFAR-100, SVHN, STL, and ImageNet are conducted to verify our theories and theory-inspired methods in [C1]-[C3].
>
> We think our work should not be judged only on what method we use and what performance we achieve. **Our theoretical results including Theorem 1, Remark 1, Remark 2, Remark 3, and Theorem 2, are original, and should not be overlooked.** We believe that the contributions, in particular [C1]-[C3], are advancements for the neural collapse area because:
>
> (1) Current neural collapse studies only focus on why neural collapse happen in balanced training [5, 8, 22, 15, 47, 24, 37], while we are the first to show that neural collapse can also be a global optimality in imbalanced training;
>
> (2) Compared with current neural collapse studies, we not only show the neural collapse global optimality, but also inspire a new loss function whose benefit is provable;
>
> (3) There are only empirical experiments in these studies showing the convergence to neural collapse, while we additionally show practical applicability of our neural collapse inspired methods by long-tailed experiments on multiple datasets including ImageNet.
>
> **Based on the statement above, we hope that our contributions can be properly recognized.**
>
> ----
> References
>
> [5] C. Fang, H. He, Q. Long, and W. J. Su. Exploring deep neural networks via layer-peeled model: Minority collapse in imbalanced training. Proceedings of the National Academy of Sciences, 118(43), 2021.
>
> [8] F. Graf, C. Hofer, M. Niethammer, and R. Kwitt. Dissecting supervised constrastive learning. In ICML, pages 3821–3830. PMLR, 2021.
>
> [15] W. Ji, Y. Lu, Y. Zhang, Z. Deng, and W. J. Su. An unconstrained layer-peeled perspective on neural collapse. In ICLR, 2022.
>
> [22] J. Lu and S. Steinerberger. Neural collapse with cross-entropy loss. arXiv preprint arXiv:2012.08465, 2020.
>
> [24] D. G. Mixon, H. Parshall, and J. Pi. Neural collapse with unconstrained features. arXiv preprint, arXiv:2011.11619, 2020.
>
> [28] V. Papyan, X. Han, and D. L. Donoho. Prevalence of neural collapse during the terminal phase of deep learning training. Proceedings of the National Academy of Sciences, 117(40):24652–24663, 2020.
>
> [37] T. Tirer and J. Bruna. Extended unconstrained features model for exploring deep neural collapse. ICML, 2022.
>
> [47] Z. Zhu, T. DING, J. Zhou, X. Li, C. You, J. Sulam, and Q. Qu. A geometric analysis of neural collapse with unconstrained features. In NeurIPS, 2021.

---

> > ### Author Response · Authors · 2022-08-02
> > **Response to your review (Part 2)**
> >
> > + In the context of learning in general, it is obviously not needed to learn a linear classifier at the end of DNN.
> >
> > Note that the "linear classifier" is not the encoding of labels. It refers to the linear mapping from the high-dimension feature (output from a backbone network) in $\\mathbb{R}^d$ to the label encoding $\\mathbb{R}^C$, where $C$ is the number of classes. So, it is a matrix in $\\mathbb{R}^{d\\times C}$.
> >
> >
> > In the general learning, such as training with the on-hot encoding and cross entropy (CE) loss, why do you claim a linear classifier is obviously not needed to learn? A learnable classifier is necessary to induce enough margin among prototypes (classifier vectors) of different classes. If a classifier is randomly fixed without learning, its prototypes may be close, so the classifier will output similar logits and be unable to classify a feature. We claim that a learnable classifier is not needed only when we fix it as an ETF, which corresponds to the largest pair-wise separation among prototypes. As described by neural collapse, it is the optimality of a learnable classifier in balanced training. Our paper shows that fixing the classifier as an ETF actually brings benefit, i.e., neural collapse can inherently happen even in imbalanced training.
> >
> > + When using an LPM model, we may get rid of learning the classifier weights. In an end-to-end learning, does not one still need to learn a method that maps features into the vertices of the ETF?
> >
> > LPM is an analytical framework where the backbone network is dropped and features are independent learnable variables. It shares similar learning behaviors to the general learning with a backbone, but facilitates theoretical analysis. A detailed comparison between LPM and general learning will be offered in our response to your later question (other comment). Prior studies on neural collapse using LPM also have learnable classifiers. We prove the benefit of fixing the classifier as an ETF, i.e., neural collapse can inherently happen even in imbalanced training (Theorem 1).
> >
> > In the general end-to-end learning with a backbone, we can still fix the classifier as an ETF and do not learn it. LPM is used just for theoretical analysis, as adopted in most neural collapse studies. But in practical implementation, a backbone is still needed to have features dependent on the input samples. As described in Lines 151 – 155 in our paper, the difference is that LPM uses the gradient $\\frac{\\partial\\mathcal{L}}{\\partial\\mathbf{h}}$ to update the variables $\mathbf{h}$, while practical general learning uses the gradient $\\frac{\\partial\\mathcal{L}}{\\partial\\mathbf{H}}$ to update the backbone parameters by multiplying with the Jacobian matrix, $\\frac{\\partial \\mathbf{H}}{\\partial \\mathbf{W}_{1:L-1}}\\frac{\\partial \\mathcal{L}}{\\partial \\mathbf{H}}$, where $\mathbf{H}$ is the collection of features and $\\mathbf{W}_\{1:L-1\}$ denotes the backbone parameters.
> >
> > Our experimental results with “ETF classifier” all have a learnable backbone and a fixed classifier as ETF in an end-to-end training manner.
> >
> >
> > + Why not consider other structures if we do not assume any CE loss? Why not a regression into the columns of an arbitrary orthogonal matrix, and there is no need to use a softmax?
> >
> > Yes, they are also feasible. Note that ETF structure has the largest pair-wise separation of $K$ prototypes in $\\mathbb{R}^d$, i.e., $\forall i\ne j, w_i^Tw_j=-\frac{1}{K-1}$ (an obtuse angle), while in an orthogonal matrix, the inner product $w_i^Tw_j=0$. Besides, the ETF structure also corresponds to the optimal Fisher discriminant ratio, with the within-class variance minimized and the between-class variance maximized. Although using an orthogonal matrix as the classifier is also feasible and has been adopted in some applications, our theoretical work is conducted in the context of neural collapse, which is mainly concerned with the ETF structure as the optimal feature-classifier alignment. So, we did not consider other structures such as orthogonal matrix.
> >
> >
> > If we use an ETF or an orthogonal matrix as the fixed classifier, we can indeed regress the feature into its corresponding prototype. It is similar to what we do in Section 4.3 (dot-regression loss) in our paper. We regress the product of feature and the prototype, instead of the feature itself. That is because if we directly regress the feature as the loss function, its gradient term of prototype will not attenuate as the optimization approaches to the optimality, like the coefficient of $(1-\\cos\\angle(\\mathbf{h},\\mathbf{w}_c^*))$ in Eq. (15) and $(1-p_c(\\mathbf{h}))$ in Eq. (12). But they have the same optimality, i.e. all features are pulled into their corresponding fixed prototypes. In our dot-regression loss proposed in Section 4,3, there is indeed no use of a softmax function.

---

> > > ### Author Response · Authors · 2022-08-02
> > > **Response to your review (Part 3)**
> > >
> > > + There are alternative methods for dealing with the problem of imbalanced datasets. The paper appears to have not considered or compared to such alternatives.
> > >
> > > Indeed, there have been a lot of methods for long-tailed classification, including re-sampling methods, re-weighting methods, and different kinds of loss function variants, such as focal loss. However, our work mainly focuses on neural collapse in imbalanced training. It is a neural collapse study whose contributions are summarized in [C1]-[C4]. We do not aim to achieve or push the state-of-the-art performance, so did no consider the advanced methods that have strong performances in long-tail classification. Some of these advanced methods are designed empirically and lack rigorous support. In contrast, the theoretical results of our work, including the neural collapse optimality in imbalanced training and the advantage of our dot-regression loss, are provable (Appendices A and B). We believe that interpretability is also an important factor that should be recognized.
> > >
> > > In our experiments, we have compared our method with the CE loss weighted by class distribution, denoted as “Learnable Classifier + CE$^*$” in Table 1. We additionally compare our method with focal loss ($\\alpha=0.25, \\gamma=2$) and class-balanced (CB) loss [48] here. The results of long-tailed classification on CIFAR-10 with Mixup and different imbalance ratios are shown as follow:
> > >
> > > | Imbalance ratio | 0.005 | 0.01 | 0.02 |
> > > | --- | --- | --- | --- |
> > > |Learnable Classifier + Weighted CE Loss | 68.5$\\pm$0.3 | 73.9$\\pm$0.3 |79.3$\\pm$0.2|
> > > |Learnable Classifier + Focal Loss | 69.9$\\pm$0.2 | 75.4$\\pm$0.3 | 78.9$\\pm$0.2 |
> > > |Learnable Classifier + CB Loss [48] | 69.3$\\pm$0.4 | 76.0$\\pm$0.2 | 79.5$\\pm$0.4 |
> > > |ETF Classifier + DR Loss | 71.9$\\pm$0.3 | 76.5$\\pm$0.3 | 81.0$\\pm$0.2 |
> > >
> > > We observe that our method still performs better than the methods that use re-weighting strategies.
> > >
> > >
> > >
> > > + Experiments in settings when there are large number of classes as in ImageNet can make the analysis significantly better. If in such settings a cross entropy loss provides better push and pull gradients than the proposed linear alternative Eq. (14).
> > >
> > > Actually, we did conduct experiments on ImageNet and showed the results in Table 3. ImageNet-LT dataset is the standard long-tailed version of ImageNet and is widely used in long-tailed classification studies. It has the same number of classes as the original ImageNet.
> > >
> > > As shown in Table 3, the superiority of our method is more remarkable when training for less epochs. It can be explained by the fact that our method directly has the classifier in its optimality and optimizes the features towards the neural collapse solution, while the learnable classifier with the CE loss needs a sufficient training process to separate classifier vectors of different classes. So, our method can be preferred when fast convergence or limited training time is required. The accuracy curves in training for the results in Table 3 are shown in Figure 6 in Appendix E.
> > >
> > > Note that our dot-regression loss in Eq. (14) is quadratic but not linear.
> > > As shown in Remark 3 in our paper, for a learnable classifier, the "pull-push" mechanism of the CE loss makes features of the same class contracted and features of different classes separated. But when the classifier has been fixed as an ETF, we only need the "pull" gradient because it is always accurate towards the optimality. So, we develop our dot-regression (DR) loss. As shown in Theorem 2, DR loss theoretically has a better convergence. Note that the proof of Theorem 2 is agnostic of the number of classes. So, no matter how many classes, our method enjoys the benefit. It also gets verified in Table 3 by experiments on ImageNet-LT that has a large number of classes.
> > >
> > > ----
> > > References
> > >
> > > [48] Cui et al., Class-Balanced Loss Based on Effective Number of Samples, CVPR 2019.

---

> > > > ### Author Response · Authors · 2022-08-02
> > > > **Response to your review (Part 4)**
> > > >
> > > > + In LPM, how are the features produced? How do you associate
> > > > the features to be from the data x?
> > > >
> > > > Neural network is highly non-convex and approximations are always necessary to perform theoretical analysis. LPM just serves as such analytical tool. It drops the backbone network and features are independent learnable variables. We did not write $\mathbf{h}_{\theta}(x)$ because $\\mathbf{h}$ is not conditioned on $x$ in LPM. Although it cannot be used to extract feature for application, it has very similar learning behaviors to the general learning in a real neural network. It is widely adopted in neural collapse studies [5, 8, 22, 15, 47, 24, 37] to facilitate theoretical analysis. We compare LPM with the general learning in a real network as follow:
> > > >
> > > > || Layer-peeled Model | Real Neural Network|
> > > > |---|---|---|
> > > > | Variables | $\\mathbf{H}$, $\\mathbf{W}$ | $\\mathbf{W}_{1:L-1}$, $\mathbf{W}$
> > > > | Gradient | $\\frac{\\partial \\mathcal{L}}{\\partial \\mathbf{H}}$, $\\frac{\\partial \\mathcal{L}}{\\partial \\mathbf{W}}$ | $\\frac{\\partial \\mathbf{H}}{\\partial \\mathbf{W}_{1:L-1}}\\cdot\\frac{\\partial \\mathcal{L}}{\\partial \\mathbf{H}}$, $\\frac{\\partial \\mathcal{L}}{\\partial \\mathbf{W}}$|
> > > >
> > > > where $\\mathbf{H}$ denotes the features, $\\mathbf{W}$ denotes the classifier, and $\\mathbf{W}_{1:L-1}$ denotes the parameters in the backbone network.
> > > >
> > > > So, we only use LPM for theoretical analysis, but still need to train a real neural network with a backbone in experiments.
> > > >
> > > >
> > > > + The paper does not describe any limitations.
> > > >
> > > > We did discuss the limitations of our work in Appendix F.
> > > >
> > > > ----
> > > > References
> > > >
> > > > [5] C. Fang, H. He, Q. Long, and W. J. Su. Exploring deep neural networks via layer-peeled model: Minority collapse in imbalanced training. Proceedings of the National Academy of Sciences, 118(43), 2021.
> > > >
> > > > [8] F. Graf, C. Hofer, M. Niethammer, and R. Kwitt. Dissecting supervised constrastive learning. In ICML, pages 3821–3830. PMLR, 2021.
> > > >
> > > > [15] W. Ji, Y. Lu, Y. Zhang, Z. Deng, and W. J. Su. An unconstrained layer-peeled perspective on neural collapse. In ICLR, 2022.
> > > >
> > > > [22] J. Lu and S. Steinerberger. Neural collapse with cross-entropy loss. arXiv preprint arXiv:2012.08465, 2020.
> > > >
> > > > [24] D. G. Mixon, H. Parshall, and J. Pi. Neural collapse with unconstrained features. arXiv preprint, arXiv:2011.11619, 2020.
> > > >
> > > > [37] T. Tirer and J. Bruna. Extended unconstrained features model for exploring deep neural collapse. ICML, 2022.
> > > >
> > > > [47] Z. Zhu, T. DING, J. Zhou, X. Li, C. You, J. Sulam, and Q. Qu. A geometric analysis of neural collapse with unconstrained features. In NeurIPS, 2021.

---

### Meta-Review · Area_Chair_raWx · 2022-08-27

**Recommendation:** Accept
**Confidence:** Certain

**Metareview:**

This paper examines the use of a random equiangular tight frame (ETF) as a replacement mechanism for the final classification layer in a deep neural network, and demonstrates experimental advantages in class-imbalanced training scenarios.

Reviewers gave drastically different assessments of this paper, with ratings ranging from reject to weak accept.  The authors provided extensive responses to all reviewers, and Reviewer amWi participated in an extended discussion with the authors.  Author responses directly addressing concerns raised by other reviewers, such as pointing to ImageNet results in response to Reviewer Nj4c asking for such experiments, appear not to have received subsequent engagement from reviewers.

The Area Chair has taken an detailed look at the paper and the entirety of the discussion, and agrees with Reviewer amWi's assessment.  The work provides an interesting examination of ETF as a novel mechanism to address class imbalanced training; the contributions meet the bar for acceptance to NeurIPS.

Reviewer amWi makes several suggestions regarding presentation of the main contributions as well as additional papers for citation and discussion, which the authors may want to take into consideration when preparing the final version of the paper.

**Award:**

No

---

### Decision · Program_Chairs · 2022-09-14

Accept